# Disentangling the genetic basis of rhizosphere microbiome assembly in tomato

Ben O. Oyserman [1,2]✉, Stalin Sarango Flores[1,3], Thom Griffioen [1], Xinya Pan[1], Elmar van der Wijk [2], Lotte Pronk [2], Wouter Lokhorst [1], Azkia Nurfikari [1], Joseph N. Paulson [4], Mercedeh Movassagh[5,6], Nejc Stopnisek [1], Anne Kupczok [2], Viviane Cordovez[1], Víctor J. Carrión [1,3], Wilco Ligterink [7], Basten L. Snoek [8], Marnix H. Medema [2,3] & Jos M. Raaijmakers [1,3]✉

Microbiomes play a pivotal role in plant growth and health, but the genetic factors involved in microbiome assembly remain largely elusive. Here, we map the molecular features of the rhizosphere microbiome as quantitative traits of a diverse hybrid population of wild and domesticated tomato. Gene content analysis of prioritized tomato quantitative trait loci suggests a genetic basis for differential recruitment of various rhizobacterial lineages, including a *Streptomyces*-associated 6.31 Mbp region harboring tomato domestication sweeps and encoding, among others, the iron regulator FIT and the water channel aquaporin SlTIP2.3. Within metagenome-assembled genomes of root-associated *Streptomyces* and *Cellvibrio*, we identify bacterial genes involved in metabolism of plant polysaccharides, iron, sulfur, trehalose, and vitamins, whose genetic variation associates with specific tomato QTLs. By integrating 'microbiomics' and quantitative plant genetics, we pinpoint putative plant and reciprocal rhizobacterial traits underlying microbiome assembly, thereby providing a first step towards plant-microbiome breeding programs.

[1] Department of Microbial Ecology, Netherlands Institute of Ecology, Wageningen, The Netherlands. [2] Bioinformatics Group, Wageningen University, Wageningen, The Netherlands. [3] Institute of Biology, Leiden University, Leiden, The Netherlands. [4] Department of Data Sciences, Genentech, Inc. South San Francisco, South San Francisco, CA, USA. [5] Department of Biostatistics, Harvard T.H. Chan School of Public Health, Boston, MA, USA. [6] Department of Data Sciences Dana Farber Cancer Institute, Harvard T.H. Chan School of Public Health, Boston, MA, USA. [7] Wageningen Seed Lab, Laboratory of Plant Physiology, Wageningen University, Wageningen, The Netherlands. [8] Theoretical Biology and Bioinformatics, Utrecht University, Utrecht, The Netherlands. ✉email: benoyserman@gmail.com; j.raaijmakers@nioo.knaw.nl

Root and shoot microbiomes are fundamental to plant growth and plant tolerance to (a)biotic stress factors. The outcome of these beneficial interactions is the emergence of specific microbiome-associated phenotypes (MAPs)[1], such as drought resilience[2], disease resistance[3], development[4], and heterosis (i.e., hybrid vigor)[5]. The microbes inhabiting the surface or internal tissues of plant roots are selectively nurtured by diverse plant-derived compounds in the form of primary and secondary metabolites[6,7]. Microbes reciprocate by supporting plant growth and producing metabolites that mediate processes such as nutrient acquisition and pathogen suppression[8,9]. Developing a blueprint of the genetic architecture for this 'chemical dialog' and how these interactions lead to specific MAPs is one of the key focal points in current plant microbiome research. The promise is that these genomic and chemical blueprints can be integrated into crop breeding programs for a new generation of 'microbiome-assisted' crops that can rely, at least in part, on specific members of the microbiome for stress protection, enhanced growth, and higher yields[10].

Selective breeding for yield-related traits has left a considerable impact on the taxonomic and functional composition of modern crop microbiomes[11,12]. Wild plant relatives represent a 'living library' of diverse genetic traits that may have been lost during domestication[13]. For example, recombinant inbred lines (RILs) of crosses between wild tomato relatives and modern tomato cultivars have been used to identify genetic loci controlling important agronomic traits, including tolerance to abiotic[14] and biotic stress[15], as well as nutritional quality and flavor profiles[16]. To date, microbiome traits are not yet considered for breeding purposes, except for specific quantitative MAPs such as the number of nodules in legume-rhizobia symbioses[17]. However, technological advances in sequencing now make it feasible to treat microbiomes as quantitative traits for selection. Quantitative approaches to map the microbiome as a phenotype have been adopted to investigate the phyllosphere microbiome and, recently, for the *Arabidopsis* and sorghum rhizosphere microbiomes[18,19]. However, actualizing microbiome features into breeding programs at a scale for crop improvement has not yet been realized. In fact, for most plant species, investigations leveraging diverse plant populations to map microbiome-associated quantitative trait loci (QTL) are still in their infancy[18–20]. In these recent studies, the microbiomes were characterized by amplicon sequencing to detect loci involved in alpha and beta diversity as well as individual OTU abundances[21]. These studies provide strong evidence that microbiome recruitment has a genetic component, but the functional nature of the corresponding plant–microbe interactions cannot be reliably elucidated from amplicon data. Hence, functional genomic features of the microbiome, as well as intraspecific diversity within microbial species, have not yet been taken into account in QTL analyses[22].

Here, we use both amplicon and shotgun metagenome sequencing to generate taxonomic as well as functional microbiome features as quantitative traits. Using an extensive RIL population of a cross between modern *Solanum lycopersicum* var. Moneymaker and wild *Solanum pimpinellifolium*[23], we identify reciprocal associations between specific plant and microbiome traits and infer putative mechanisms for rhizosphere microbiome assembly. Using the modern allele as a reference, we find QTLs for numerous taxonomic and metagenomic features of the microbiome with both positive and negative effects. We observe more positive effects related to increases in microbiome feature abundance for the modern reference allele compared to the wild reference allele, suggesting that domestication has had a significant impact on rhizosphere microbiome assembly. We identify plant traits related to growth, stress, amino acid metabolism, iron and water acquisition, hormonal responses, and terpene biosynthesis, whereas the microbial traits we identify are related to the metabolism of plant cell wall polysaccharides, vitamins, sulfur, and iron. Furthermore, we show that amplicon-based approaches allow detection of QTLs for rarer microbial taxa, whereas shotgun metagenomics allowed mapping to smaller and thus more defined plant genomic regions. Together, these results demonstrate the power of an integrated approach to disentangle and prioritize specific genomic regions and genes in both plants and microbes associated with microbiome assembly.

## Results

**Baseline analyses of the tomato recombinant inbred line population**. Prior to detailed metagenome analyses of the microbiome of the tomato RIL population, we first investigated whether QTLs previously identified in the same RIL population under sterile in vitro conditions could be replicated in our experiment conducted under greenhouse conditions with a commercial tomato greenhouse soil (Fig. 1a, b and Supplementary Data 1)[24]. We identified QTLs for shoot dry weight (SDW) coinciding with a QTL identified previously on chromosome 9[24]. Similarly, we identified QTLs for rhizosphere mass (RM), defined here as a the total mass of the roots with tightly adhering soil, which coincides with root trait QTLs previously identified for lateral root number, fresh and dry shoot weight, lateral root density per branched zone and total root size (Fig. 1b)[24]. An analysis of variance (ANOVA) yielded significant variation in SDW based on the additivity of alleles linked to SDW (zero, one, or two alleles) ($F_{(2, 186)} = 16.02$, $p = 3.76$ e–07) (Fig. 1c, d). A post hoc Tukey test further demonstrated significant differences between all pairwise comparisons ($p < 0.05$). For RM, an ANOVA yielded a significant difference ($F_{(2, 186)} = 16.02$, $p = 3.76$ e–07); a post hoc Tukey test demonstrated a statistically significant difference only between the presence of either one or two alleles ($p < 0.05$), but did not support additivity ($p = 0.15$) (Fig. 1e, f). Collectively, our results confirm and extend earlier work conducted on the same tomato RIL population in vitro[24], providing a solid basis for QTL mapping of taxonomic and genomic features of the rhizosphere microbiome

**Taxonomic microbiome features as quantitative traits**. To investigate molecular features of the microbiome as quantitative traits, we conducted 16S rRNA gene amplicon sequencing of 225 rhizosphere samples, including unplanted bulk soil, parental tomato genotypes, and all 96 RIL accessions in duplicate (BioProject ID PRJNA787039). We observed separation between the microbiomes of rhizosphere and bulk soil, between the microbiomes of the two parental tomato genotypes, and the RIL accession microbiomes (Fig. 2a). To limit multiple testing and to focus on common microbiome features with sufficient coverage across all accessions, we prioritized the rhizosphere-enriched amplicon sequence variants (ASVs) to those present in 50% or more of the RIL accessions (Fig. 2b). A QTL analysis with these prioritized ASVs was run with R/qtl2[25] using a high-density tomato genotype map[26], harvest date, post-harvest total bulk soil mass, RM, number of leaves at harvest, and SDW as covariates.

We identified 48 QTL peaks, across 45 distinct loci, significantly associated with 33 ASVs (Supplementary Data 6). Our logarithm of the odds (LOD) thresholds for significance had been determined by pooled permutations from all ASVs to attain a genome-wide threshold of $P$ 0.05 (LOD 3.35) and $P$ 0.2 (LOD 2.64). The modern allele was set at reference, such that negative effects were relatively more associated with the wild allele and positive effects with the modern allele. Of the significant QTLs, 16 were microbiome features less abundant compared to the reference allele, whereas 32 were microbiome features more

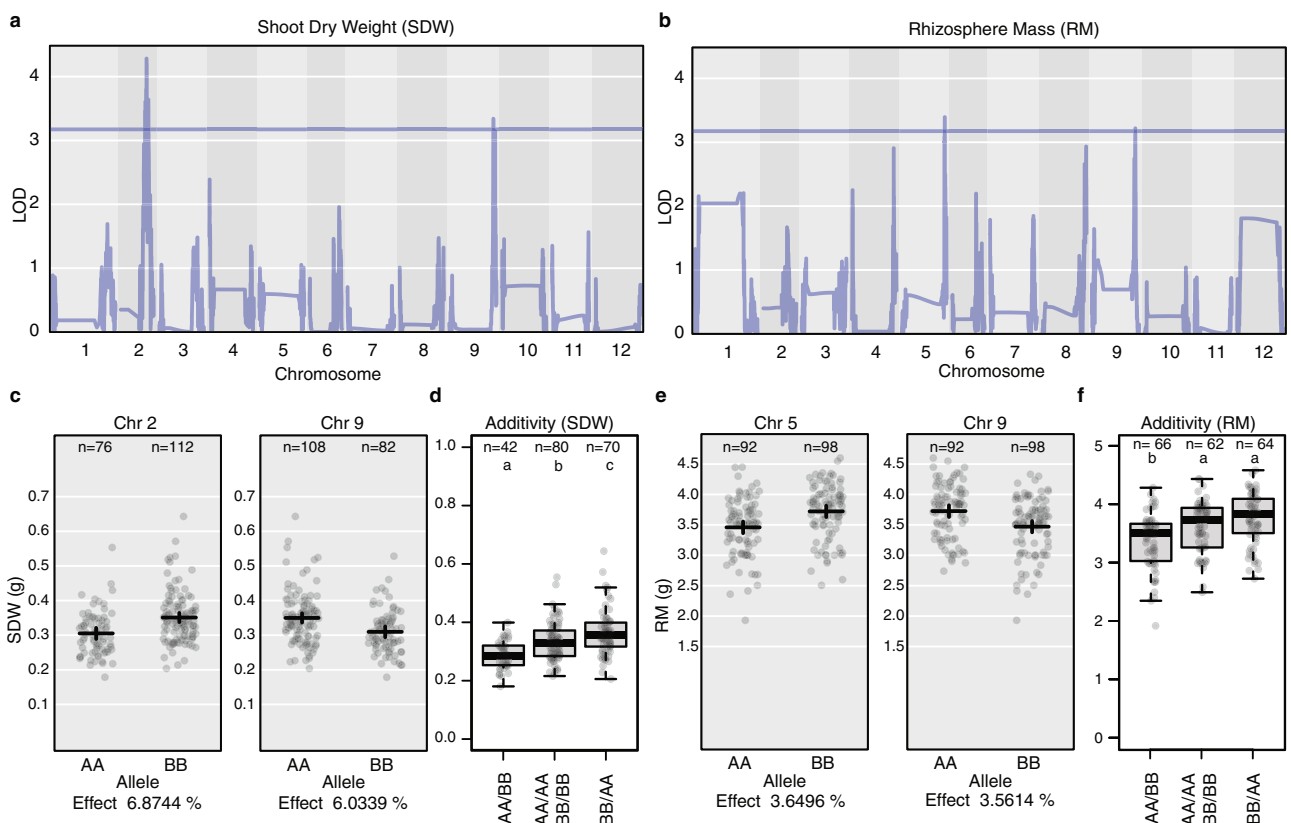

**Fig. 1 Replication of shoot dry weight and rhizosphere mass QTLs from previous studies. a** QTLs identified for SDW on chromosome 9 position 63.63719184 and chromosome 2 position 42.7291229, coinciding with a QTL identified previously (chromosome 9 position 62.897108) by Khan et al 2012. **b** QTL of RM on chromosome 5 position 62.00574891, and chromosome 9 position 62.71397636, which coincide with root trait QTLs previously identified for lateral root number chromosome 5 position 53.4–86.1, and several on chromosome 9, including fresh and dry shoot weight, (chromosome 9 position 81.3–95.3), lateral root density per branched zone (chromosome 9 position 33.8–88.7), and total root size (chromosome 9 position 39.4–75.1) from Khan et al 2021. **c** Scatter plots showing the distribution of SDW measurements on chromosome 2 position 42.7291229 and chromosome 9 position 63.63719184 for both modern (AA) and wild (BB) alleles. For the QTL on Chromosome 2, $n = 76$ and 112 biologically independent samples for AA and BB respectively. For the QTL on Chromosome 9, $n = 106$ and 82 biologically independent samples for AA and BB respectively. In addition to the scatter plot, data are presented as mean values $+/-$ two times the SEM. **d** Significant additivity of alleles for SDW ($p < 0.05$); n of 42, 80, and 70 for biologically independent plants containing neither allele (AA/BB), either BB allele on chromosome 2, or AA on chromosome 9 (AA/AA or BB/BB), or both AA and BB alleles (BB/AA) respectively. In addition to the scatter plot, data are presented with boxplots representing the median value, the interquartile range, and whiskers representing the minimal and maximal values excluding points greater than 1.5 times the interquartile range. **e** Scatter plots showing the distribution of RM measurements on chromosome 5 (pos 62.00574891), and chromosome 9 (pos 62.71397636) for both modern (AA) and wild (BB) alleles. For the QTL on Chromosome 5, $n = 92$ and 98 biologically independent samples for AA and BB respectively. For the QTL on Chromosome 9, $n = 92$ and 98 biologically independent samples for AA and BB respectively. In addition to the scatter plot, data are presented as mean values $+/-$ two times the SEM. **f** No additivity of alleles was observed for RM. In addition to the scatter plot, data are presented with boxplots representing the median value, the interquartile range, and whiskers representing the minimal and maximal values excluding points greater than 1.5 times the interquartile range. Source data are provided as a Source Data file.

abundant in presence of the modern reference allele. The QTLs on chromosomes 11, 10, 8, and 2 were associated with increases in abundance in presence of the modern reference allele. In contrast, the sole QTL on chromosome 7 was negative relative to the reference. All other chromosomes contained a mix of QTLs with positive and negative effects on ASV abundance relative to the reference allele (Fig. 3a). While many rhizobacterial lineages were linked to a single QTL (14 out of 25 unique taxonomies), others were linked to two or more QTLs (7 and 4 taxa, respectively) (Fig. 3b). Of the lineages with multiple QTLs, most were positive relative to the reference allele. One salient exception was *Methylophilaceae*, with a total of 9 QTLs that were both positive and negative relative to the reference and distributed across chromosomes 3 (positive, x2), 4 (positive), 7 (negative), 11 (positive x2), and 12 (negative x3) (Fig. 3c). Another salient feature of the QTL analysis was the hotspot for microbiome

assembly identified on chromosome 11, including a significant linkage with ASVs from *Adhaeribacter, Caulobacter, Devosia*, Rhizobiaceae, *Massilia,* and *Methylophilaceae* (Fig. 3c).

In addition to individual ASVs, we investigated diversity metrics as quantitative traits using Shannon index and principal coordinate analysis (PCoA) with Bray–Curtis dissimilarity. For each approach, we calculated diversity statistics first using all ASVs with a relative abundance greater than the effective samples size[27], and second using the rhizosphere-enriched ASVs present in 50% or more of the RIL accessions. For the Shannon index, LOD thresholds for significance were determined by permutations to attain a genome-wide threshold of P 0.05 (LOD 3.27) and P 0.2 (LOD 2.63). Two QTLs were identified on chromosomes 1 and 3 (Supplementary Figs. 1 and 2) using all, and prioritized, ASVs to calculate Shannon Diversity respectively. Of note, the QTL on chromosome 1 overlaps with the confidence interval of

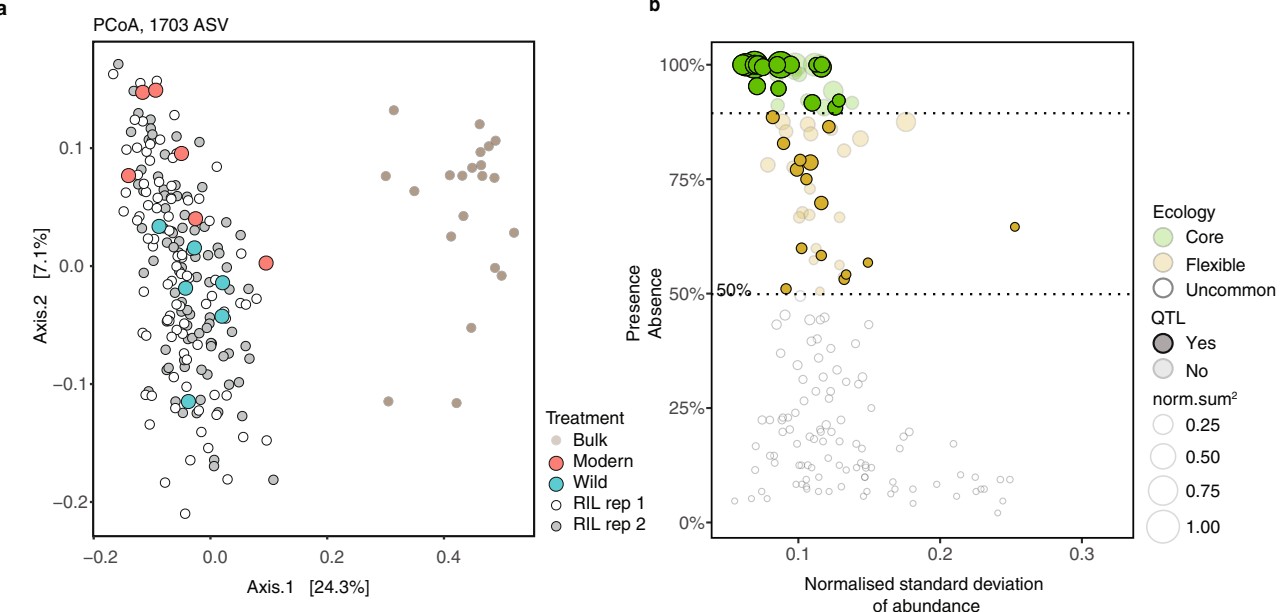

**Fig. 2 The 16S rRNA microbiomes of the bulk soils, modern and wild tomatoes, and RIL population. a** A PCoA analysis of ASVs demonstrating a separation between the bulk soil and rhizosphere microbiomes. The rhizosphere of RIL accessions distribute around the wild and modern rhizospheres. Separation between the two replicate RIL populations was not observed. **b** To limit multiple testing, a QTL analysis was conducted only on ASV that were observed in over 50% of accessions. The 33 ASV that are subsequently found with QTLs are shown with full opacity. All other ASV are partially transparent. Source data are provided as a Source Data file.

the *Cellvibrio* QTL highlighted later in the results section. For the PCoA, the first two components were mapped as quantitative traits. A LOD threshold for significance was determined by permutations to attain a genome-wide threshold of $P$ 0.05 (LOD 3.41) and $P$ 0.2 (LOD 2.71). A single QTL was identified on chromosome 6 in the same position as the QTL identified previously for *Streptomyces* ASV 5 (Supplementary Fig. 3). Of further interest is that all diversity metric QTLs were negative relative to the reference. Thus, while genetic changes during domestication may have made some ASVs more or less abundant, these genetic changes also impacted overall diversity. Given the non-independence of sequencing-based microbiome features, we suggest caution in interpreting the results of using diversity metrics as microbiome features.

Effect size is an important factor when mapping the genetic architecture of quantitative traits. While some QTLs have large effect sizes, many small effect QTLs may explain a large proportion of trait variation[28]. To date, there is little understanding of the distribution of the effect sizes of QTLs for microbiome features. Here we show that the absolute values of the effect sizes of the 48 QTLs on ASV relative abundance ranged from 1.3 to 17%, with an average effect size of approximately 5%, comparable to the effects seen for SDW and RM (Fig. 1c, e). The largest QTL effects were positive for an ASV in the genus *Qipengyuania* (17%), and an ASV in *Edaphobaculum* (10%). However, no statistical difference was found between the absolute value of positive and negative effect sizes ($p = 0.78$, two-tailed $t$-test). Furthermore, for those lineages with sufficient representation at the class level (Bacteroidia, Alphaproteobacteria, and Gammaproteobacteria), there was no statistically significant difference between effect size (F(3, 16) = 0.072, $p = 0.974$). However, an ANOVA on the positive effect size at genus level demonstrated significant differences between lineages (F(3, 16) = 12.94, $p = 1.15$ e−04). A post hoc Tukey test demonstrated QTLs for *Massilia* with a larger positive effect size than other lineages with sufficient sample size for comparison (Fig. 3d). Collectively, our amplicon analysis provided a broad picture,

suggesting that the assembly of bacteria in the tomato rhizosphere is a complex trait governed by a combination of multiple loci, some being ASV specific, some being pleiotropic for different ASVs, and with heterogenous effect sizes on ASV abundance (Fig. 3d). While QTLs were identified with both positive and negative effects relative to the reference modern allele, the large number of positive effects suggests domestication impacted rhizosphere microbiome assembly.

**Functional microbiome features as quantitative traits**. To understand the functional traits associated with rhizosphere microbiome assembly, we generated shotgun metagenomes for the rhizosphere microbiome of each accession in the tomato RIL population (96 total), as well as six samples of the modern tomato parent, five samples of the wild tomato parent and seven bulk soil samples (BioProject ID PRJNA789467). After pre-processing, a co-assembly strategy using all metagenomes was implemented (see Supplementary Methods section 4.2.2 for more detail). Subsequently, bin and contig abundances were determined by read depth using CSS normalization, a computational method to adjust for compositional bias[27]. QTL mapping was conducted for the rhizosphere-enriched contig and bin abundances. A PCoA analysis of the contigs demonstrated separation between the bulk soil and RIL rhizosphere microbiomes (Supplementary Fig. 9). Binning was done using Metabat2 (version 2:2.15)[29] and genomic quality of the output was evaluated by CheckM[30] (Supplementary Data 7). The bins and assembled contigs larger than 10 kb are publicly available (https://doi.org/10.5281/zenodo.6561541). All contigs of 10 kb and larger were taxonomically assigned using Kraken[31] (Supplementary Data 8). With nearly 40 million contigs being assembled, the effects of multiple testing were reduced by prioritizing rhizosphere-enriched contigs (relative to the bulk soil) which were larger than 10 kb and with an enrichment greater than 4-fold. After these stringent prioritization steps, 1249 contigs were remaining. The functional potential of these rhizosphere-enriched contigs represented 8.3% of protein clusters identified in

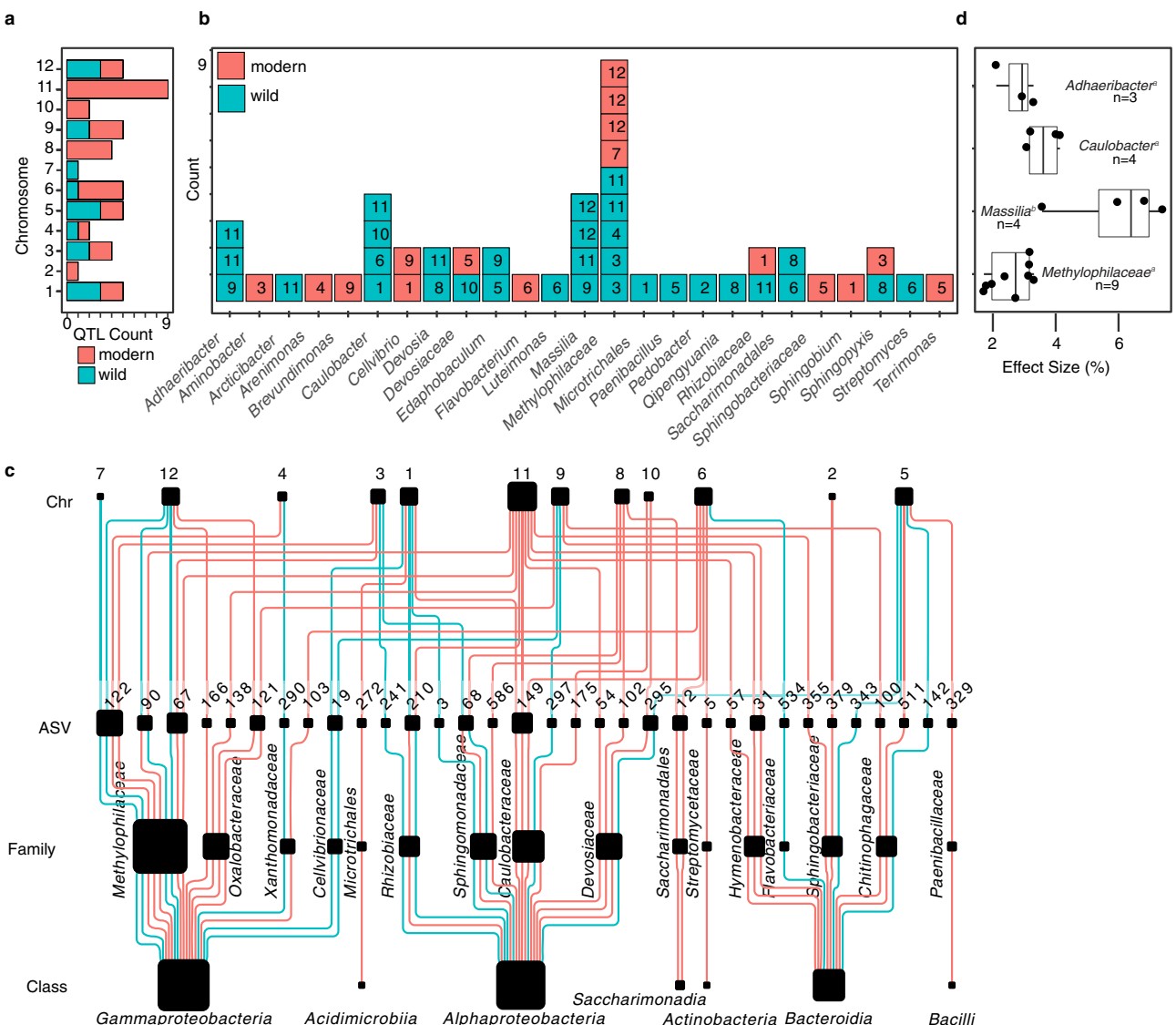

**Fig. 3 The 16S rRNA QTLs. a** A color coded summary of the number of 16S rRNA QTLs identified per chromosome to wild and modern alleles. **b** A summary of the number of 16S rRNA QTLs found by taxonomies, with the chromosome of each QTL represented within each square. The presence and absence of dark borders around each square are used to indicate a QTL linked to higher abundance for a wild allele and modern allele respectively. **c** A hierarchically structured network depicting the 16S rRNA QTLs identified in this study. From the top to bottom: the nodes in the first row represent tomato chromosomes, which are linked to specific ASV in the next row, which are linked to different families and classes of bacteria in subsequent rows. The size of the chromosome nodes is weighted by the number of outbound edges. The ASV, family, and class node sizes are weighted by the number of in-bound edges. The edges are color coded based on negative effect relative to the modern reference (e.g., wild allele), and positive effect relative to the modern reference (e.g., modern allele). The abundance of individual ASV, and at different taxonomic levels, is determined through a complex interaction of multiple alleles from both modern and wild origin. **d** A statistical analysis of the four lineages with 3 or greater QTLs shows that the absolute value of effect size for different lineages is different. Specifically, we find that the effect size for ASV within *Massila* (n = 4) was significantly larger than for the other lineages (*Adhaeribacter*, n = 3; *Caulobacter*, n = 4; *Methylophylaceae*, n = 9). The effect size was calculated as the percent change relative to the mean CSS abundance for each ASV. In addition to the scatter plot, data are presented with boxplots representing the median value, the interquartile range, and whiskers representing the minimal and maximal values excluding points greater than 1.5 times the interquartile range. Source data are provided as a Source Data file.

all contigs greater than 10 kb by MMseqs2 using a 50% protein identity threshold[32]. Approximately 25% of all proteins were contained within these clusters, suggesting that a considerable fraction of functional diversity was maintained during the prioritization. Only bins with greater than 90% completion and less than 5% contamination were mapped (33 out of 588 bins). As with the ASVs, harvest date, bulk soil mass, RM, number of leaves at harvest, and SDW were used as covariates in QTL mapping.

We identified 7 significant bin QTLs (LOD > 3.40, P < 0.05) (Supplementary Data 9) including *Streptomyces* bin 72 with a

positive effect on tomato chromosomes 6 and 11. For the contigs, a total of 717 QTLs at 26 unique positions on tomato chromosomes 1, 4, 5, 6, 9, and 11 were identified (Supplementary Data 10), corresponding to 476 metagenomic contigs from 10 different genera (LOD > 3.47, P < 0.05). The largest number of contig QTLs were linked to the *Streptomyces*, *Cellvibrio,* and *Sphingopyxis* lineages (Fig. 4a). The *Streptomyces* contigs mapped to QTLs on tomato chromosomes 4 (46 contigs, negative), 6 (190 contigs, positive), and 11 (257 contigs, positive), with a subset of contigs mapping to two or all three of these positions (Fig. 4b).

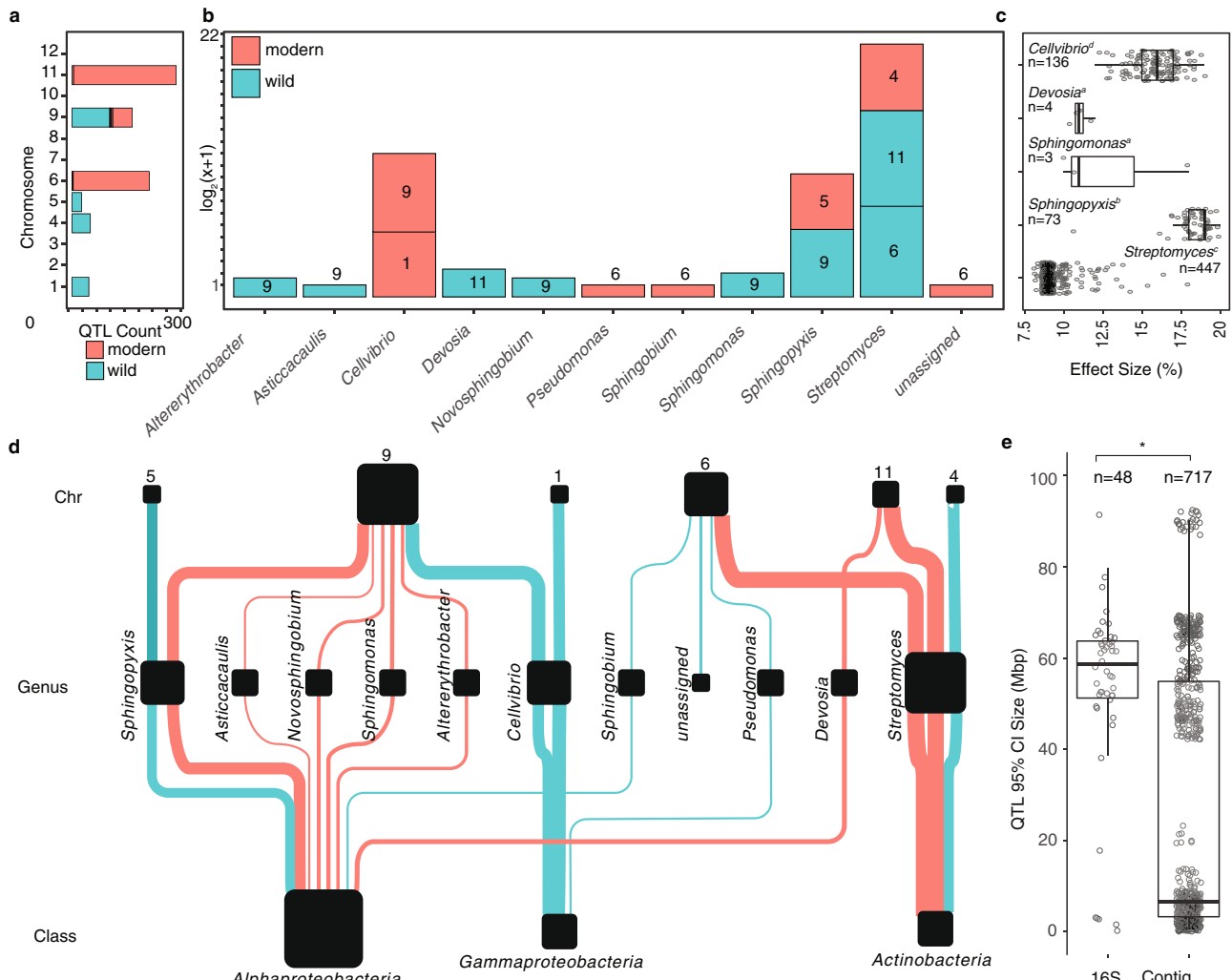

**Fig. 4 The contig QTLs. a** A color coded summary of the number of contig QTLs identified per chromosome to wild and modern alleles. **b** A summary of the number of contig QTLs found by taxonomies, with the chromosome of each QTL represented within each square. The presence and absence of dark borders around each square are used to indicate a QTL linked to higher abundance for a wild allele and modern allele respectively. **c** The effect sizes for contigs from each lineage *Cellvibrio* (n = 136), *Devosia* (n = 4), *Sphingomonas* (n = 3), *Sphingopyxis* (n = 73) and *Streptomyces* (n = 447) were significantly different as indicated by letters (F(14, 702) = 530.9 *p* < 2e−16). In addition to the scatter plot, data are presented with boxplots representing the median value, the interquartile range, and whiskers representing the minimal and maximal values excluding points greater than 1.5 times the interquartile range. **d** A hierarchically structured network depicting the contig rRNA QTLs identified in this study. From the top to bottom rows are the tomato chromosomes, which are linked to specific contigs, which are linked to different families and classes of bacteria. The size of the chromosome nodes is weighted by the number of outbound edges. The ASV, family, and class node sizes are weighted by the number of in-bound edges. In addition to the scatter plot, data are presented with boxplots representing the median value, the interquartile range, and whiskers representing the minimal and maximal values excluding points greater than 1.5 times the interquartile range. **e** When comparing the 95% confidence interval of 16S rRNA amplicon QTLs (n = 48) and contig QTLs (n = 717), the 95% confidence interval of contig QTLs was significantly smaller (two-sided *t*-test, *p* = 3.32E−09). In addition to the scatter plot, data are presented with boxplots representing the median value, the interquartile range, and whiskers representing the minimal and maximal values excluding points greater than 1.5 times the interquartile range. Source data are provided as a Source Data file.

These findings corroborate and expand upon the *Streptomyces* QTL identified on chromosome 6 using our 16S rRNA gene amplicon data, as well as that of the bin QTLs identified on chromosomes 6 and 11. The *Cellvibrio* contigs mapped to chromosome 1 (42 contigs, negative) and chromosome 9 (94 contigs, negative), again corroborating the findings from our 16S rRNA gene amplicon analysis described above. In contrast, the *Sphingopyxis* QTLs identified on chromosome 5 (24 contigs, negative) and 9 (49 contigs, positive) did not correspond to the QTLs identified on chromosomes 8 and 3 in the 16S rRNA gene amplicon analysis. Four contigs for *Devosia* also corroborated the results of the 16S QTL analysis. The effect sizes ranged from 9 to 21% and were significantly different (F(14, 702) = 530.9 *p* < 2e

−16) between QTL and lineages (Fig. 4c). As with the 16S rRNA amplicon analysis, some of the highest LOD scores were for *Devosia*. Also, the effect size of the *Sphingopyxis* contigs was large (±20% on average), above 15% for *Cellvibrio*, and approximately 10% for *Streptomyces*. The average QTL region was 51.59 Mbps for the 16S rRNA gene amplicon sequences and 26.64 Mbps for the metagenomic contigs (two-sided *t*-test, *p* = 3.32E−09) (Fig. 4e). A more striking contrast was observed in the difference between the median size of amplicon and contig QTL regions which were 58.56 Mbp and only 6.47 Mbp, respectively. In summary, while many more taxa were identified in the amplicon-based QTL analysis, the metagenome-based QTL analysis provided QTLs with much smaller confidence intervals (Fig. 4e).

 

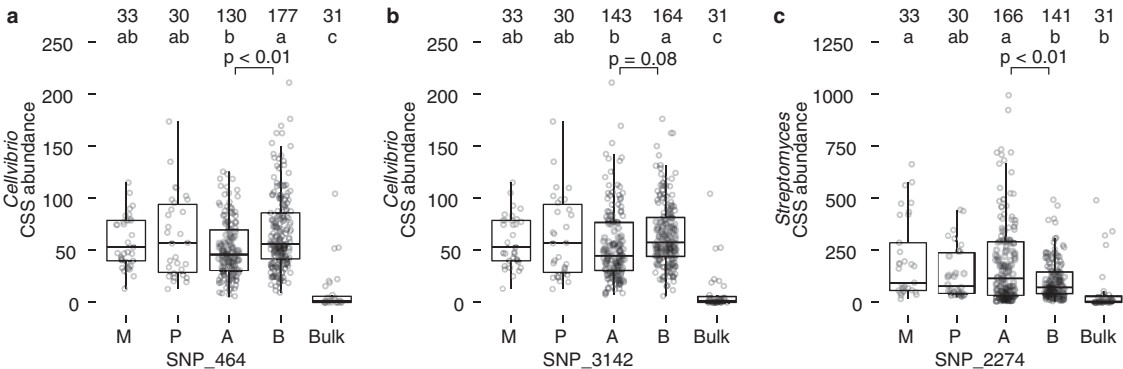

**Fig. 5 Validation of *Cellvibrio* and *Streptomyces* 16S rRNA QTLs with bulk segregant analysis.** A total of 77 RIL accessions were grown with approximately four biological replicates per accession, as well as 33 modern, 30 wild and 31 bulk samples (see Supplementary Data 13). The number of replicates representing for each treatment is detailed in the top row of each panel. The number of replicates within the RIL population is represented by either an A (modern) or B (wild) allele, which depends on the marker in question. The row below represents the statistical group based on Tukey's HSD. In addition to the scatter plot, data are presented with boxplots representing the median value, the interquartile range, and whiskers representing the minimal and maximal values excluding points greater than 1.5 times the interquartile range. **a** The CSS normalized abundances of *Cellvibrio* 16S rRNA in bulk soil (B), modern (M), wild (W), and RIL accessions at marker position 464 on chromosome 1. At this position, 32 and 45 RIL accessions with modern (A) and wild alleles (B) were used (130 and 177 samples with biological replication respectively). ANOVA showed a statistical difference between genotypes and bulk soil ($F(4, 396) = 21.56$, $p = 4.16\ e{-}16$), A post hoc Tukey test supported the conclusion that wild allele at markers 464 associated with increased abundance *Cellvibrio* ($p = 3.913\ e{-}04$). **b** Similarly, for marker 3142 on chromosome 9, there were a total of 35 and 42 RIL accessions with modern (A) and wild alleles (B), (143 and 164 samples with biological replication respectively). ANOVA showed a statistical difference between genotypes and bulk soil ($F(4, 396) = 18.43$, $p = 6.68\ e{-}14$), A post hoc Tukey HSD test supported the conclusion that wild allele at markers 464 associated with increased abundance *Cellvibrio* ($p = 0.08$). **c** The normalized CSS abundances of *Streptomyces* 16S rRNA and sequences in bulk soil (B), modern (M), wild (W), and RIL accessions at marker 2274 on chromosome 6. There was a total of 42 and 35 RIL accessions with modern (A) and wild alleles (B), (166 and 141 samples with biological replication respectively). ANOVA showed a statistical difference between genotypes and bulk soil ($F(4, 396) = 8.423$, $p = 1.57\ e{-}06$), A post hoc Tukey HSD test supported the conclusion that wild allele at markers 464 associated with increased abundance *Streptomyces* ($p = 1.152\ e{-}04$). Source data are provided as a Source Data file.

**Amplicon-based bulk segregant analysis of *Streptomyces* and *Cellvibrio* abundance.** The two most abundant rhizosphere taxa with replicated patterns for amplicon and metagenome-based QTLs were *Streptomyces* and *Cellvibrio*. Therefore, we sought to provide additional independent support for these QTLs using a bulk segregant analysis of an independent population of parental and RIL genotypes (Supplementary Data 11). In particular, we tested the previously identified amplicon-based QTLs associated with higher *Cellvibrio* abundance at markers 464 and 3142 on chromosomes 1 and 9, respectively with higher *Streptomyces* abundance at marker 2274 on chromosome 6 (Fig. 5). In each case, ANOVA showed a statistical difference between genotypes and bulk soil, respectively ($F(4, 396) = 21.56$, $p = 4.16\ e{-}16$), ($F(4, 396) = 18.43$, $p = 6.68\ e{-}14$), ($F(4, 396) = 8.423$, $p = 1.57\ e{-}06$). A post hoc Tukey HSD test supported the conclusion that wild allele at markers 464 and 3142 on chromosomes 1 and 9, respectively, are indeed associated with increased abundance *Cellvibrio* ($p = 3.913\ e{-}04$, and $p = 0.08$, respectively), while the modern allele at markers 2274 on chromosome 6 was significantly associated with increased abundance of *Streptomyces* ($p = 1.152\ e{-}04$).

**Host genetics and rhizosphere microbiome assembly.** A subset of 5 regions consistent across both the amplicon and metagenome-based analyses were prioritized with an average size of 2.68 Mbps (Supplementary Data 12). These included positions on chromosome 1 (positions 87.36–90.49 Mbps), chromosome 9 (pos 62.03–63.32 Mbps), chromosome 5 (pos 61.54–63.38), chromosome 6 (pos 33.99–40.3 Mbps), and chromosome 11 (pos 53.06–53.89 Mbps). In total, 1359 genes were identified in these regions. Potential candidate genes with root-specific transcriptional patterns, defined as a 4 fold increase in the roots compared to leaf samples, were further prioritized using a publicly available

RNA-seq dataset[33]. Based on this analysis, a subset of 192 root specific plant genes were identified (Supplementary Data 13). A total of 98 root specific plant genes were linked to *Streptomyces* on chromosome 6 (84 genes) and 11 (14 genes) (Fig. 6). Intriguingly, 61 of these genes were found in regions previously identified to be subjected to selective sweeps, regions of fixed low genetic diversity, related to tomato domestication as well as to subsequent sweeps related to improvements in fruit quality[34] (Supplementary Fig. 4). While it remains unclear whether the relationship between selective sweeps and changes in microbial feature abundance is causal or coincidental; here we reveal a genomic signature that the domestication process impacted alleles involved in microbiome assembly.

Two of the most salient genes in this list included genes with high transcription in the roots; an aquaporin and a Fer-like iron deficiency-induced transcription factor (FIT). The aquaporin (SlTIP2.3) has the highest fold change of all tonoplast intrinsic proteins in tomato roots as compared to all other organs[32,33], while the FIT gene is a bHLH transcriptional regulator controlling iron homeostasis in tomato[34,35]. Other genes within this region on chromosome 6 include a glycine rich protein, a receptor-like kinase known to be upregulated during drought[36], alcohol dehydrogenase, numerous phosphatases, expansins, ethylene-responsive transcription factors, gibberellin receptors, aminocyclopropane-1-carboxylate oxidase (ACO), an enzyme involved in the last step of ethylene biosynthesis, and finally, alpha-humulene and (-)-(E)-beta-caryophyllene, a known tomato terpene and signaling molecule in tomato[37,38] and also acting as a volatile in microbiome assembly[39]. Root specific genes involved in carbohydrate, protein, and amino metabolism were also identified, including trypsin-alpha amylase inhibitor, prolyl 4-hydroxylase, polygalacturonase, trehalose phosphatase, glycogenin, xyloglucan fucosyltransferase, and a metallocarboxypeptidase inhibitor, spermidine synthase, acetolactate synthases,

 

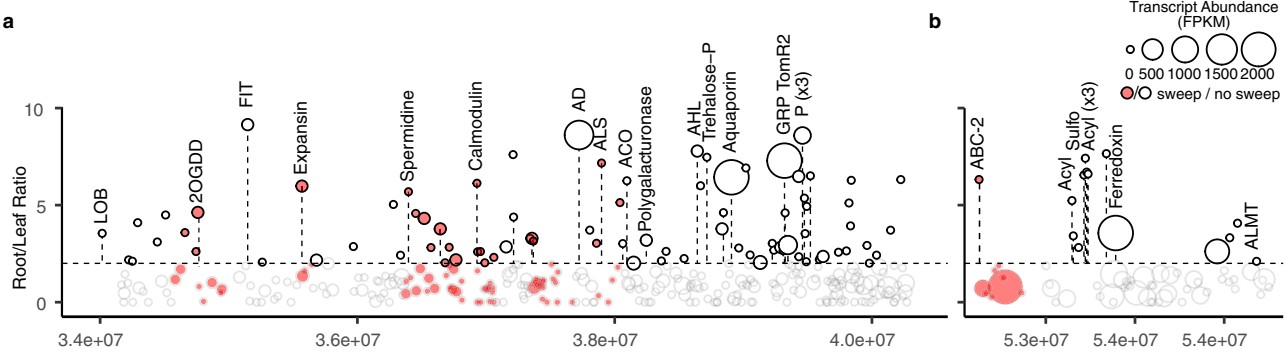

**Fig. 6 The prioritized regions of the *Streptomyces* QTL on chromosomes 6 and 11 overlaying previously reported data on transcript expression and genetic sweeps due to domestication.** Within each region, the $\log_2$ ratio gene expression patterns from leaf and root materials were calculated and those with a $\log_2$ greater than 2, as delineated by the dotted line, were further prioritized. The $\log_2$ root transcript abundances (fragments per kilobase of exon per million mapped fragments, FPKM) are depicted by the size of the bubble. Previously reported genetic sweeps are indicated in red. **a** The 6.31 Mbp region on chromosome 6 position 33.99–40.3 Mbps. Abbreviations of highlighted genes: LOB - LOB domain protein 4, 2OGDD - 2-Oxoglutarate-dependent dioxygenases, FIT - FIT (Fer-like iron deficiency-induced transcription factor), Spermidine - Spermidine synthase, AD - Alcohol dehydrogenase 2, ALS - Acetolactate synthase, ACO - 1-aminocyclopropane-1-carboxylate oxidase, Polygalacturonase, AHL - AT-hook motif nuclear-localized protein, Trehalose-P - Trehalose 6-phosphate phosphatase, Aquaporin - Tonoplast intrinsic protein 23/Aquaporin, GPR TomR2 - Glycine-rich protein TomR2, P - Acid phosphatase (×3). **b** The 0.83 Mbp region on chromosome 11 position 53.06–54.89 Mbps. Abbreviations of highlighted genes: ABC-2 - ABC-2 type transporter, Acyl–Acyltransferase (×4), Sulfo–Sulfotransferase, ALMT- Aluminum-activated malate transporter. Source data are provided as a Source Data file.

alanine aminotransferase, and an amino acid permease. On chromosome 11, a ferrodoxin, an aluminum-activated malate transporter[40], and a cluster of various acetyltransferases and a sulfotransferase were identified. An aluminum-activated malate transporter was also identified in the QTL region on chromosome 6, which has been linked to increased malate accumulation in both fruit and roots[41].

A total of 57 root specific genes were identified in the QTL regions on chromosome 1 and 9 linked to *Cellvibrio*. These include a cytochrome p450 involved in coumarin synthesis, numerous extensins, phosphatases, respiratory burst oxidase-like protein, iron chelator nicotianamine synthase[42,43], and on chromosome 11 phenazine biosynthesis. On chromosome 5, 37 root specific genes were identified including multiple peroxidases, glutamine synthetase, rhamnogalacturonate lyase, pectinesterase, metacaspase, and trehalose-phosphatase. Furthermore, numerous ethylene responsive transcription factors and receptor-like kinases were observed. The QTL on chromosome 1 contains genome-wide sweeps associated with the initial tomato domestication and subsequent improvements of fruit quality traits, suggesting that one or both of these events were connected to or act as a 'side effect' on the decreased abundance of *Cellvibrio* in the tomato rhizosphere.

**Illuminating metagenomic traits in *Cellvibrio* and *Streptomyces*.** To further investigate the potential functional importance of the 476 rhizosphere-enriched metagenomic contigs mapped as QTLs, we performed a deeper analysis into their functional gene content (Supplementary Data 14, 15, and 16). An antiSMASH[44] analysis identified 30 biosynthetic gene clusters (BGCs) across these contigs. These BGCs largely originated from contigs taxonomically assigned to *Cellvibrio* and *Streptomyces*. They included several gene clusters potentially associated with root colonization, such as two melanin BGCs (c00216, NODE_5919; c00255, NODE_7250) from *Streptomyces* (which have been positively associated with colonization[45]) and a *Cellvibrio* aryl polyene BGC (c00185, NODE_4941), which is thought to protect bacteria against reactive oxygen species generated during immune responses of the host plant[46]. The contigs also contained gene clusters potentially beneficial to the host, such as BGCs encoding

iron-scavenging siderophores, which have been associated with disease suppression in tomato[47]; specifically, homologs of coelichelin and desferrioxamine BGCs from streptomycetes were found (c00269, NODE_7969, and c00122, NODE_3362), three IucA/IucC-like putative siderophore synthetase gene clusters (c00106, NODE_2973; c00041, NODE_1131; c00238, NODE_6661), as well as a *Cellvibrio* NRPS-PKS gene cluster (c00001, NODE_101) most likely encoding the production of a siderophore based on the presence of a TonB-dependent siderophore receptor-encoding gene as well as a putative *tauD*-like siderophore amino acid β-hydroxylase-encoding gene[48]. The *Cellvibrio* contigs also contain several genes relevant for carbohydrate catabolism. For example, homologs of *xyl31a* (B2R_23365) and *bgl35a* (B2R_06825-06826) were detected (with 78%, 79 and 65% amino acid identity, respectively), genes that have been shown to be responsible for utilization of the abundant plant cell wall polysaccharide xyloglucan in *Cellvibrio japonicus*[49]. In addition, a possible homolog of the β-glucosidase gene *bgl3D*[50] (B2R_26663), involved in xyloglucan utilization, was also identified, having high similarity to *bgl3D* from *Cellvibrio japonicus* (64% amino acid identity). Also, putative cellulose-hydrolizing enzymes were detected, such as a homolog (B2R_21082) of the cellobiohydrolase *cel6A* from *Cellvibrio japonicus*[51] encoded in a complex locus of nine carbohydrate-acting enzymes annotated on this contig (NODE_5090) by DBCAN[52] (Supplementary Data 14). Collectively, these results point to a possible role of microbial traits related to iron acquisition and metabolism of plant polysaccharides in tomato rhizosphere microbiome assembly.

Contigs of the metagenome-assembled genome (MAG) associated with *Streptomyces* ASV5 (the key taxon associated with tomato QTLs described above) contained a multitude of functional genes potentially relevant for host-microbe interactions. Taxonomically, the ASV5 MAG was most closely related to a clade of streptomycetes that includes type strains of species such as *arenae*, *flavovariabilis*, *variegatus*, and *chartreusis*. To understand how tomato might differentially recruit ASV5 streptomycetes, we analyzed the MAG for genes and gene clusters potentially involved in colonization. Intriguingly, we found contigs to be rich in genes associated with plant cell wall degradation. In particular, we

identified a family 6 glycosyl hydrolases (B2R_10154) of which the glycosyl hydrolase domain has 84% amino acid identity to that of the SACTE_0237 protein that was recently shown to be essential for the high cellulolytic activity of *Streptomyces* sp. SirexAA-E[31]. Additionally, we detected a homolog (82% amino acid identity) of *Streptomyces reticuli* avicelase, a well-studied cellulase enzyme that degrades cellulose into cellobiose[53] (B2R_29198). Larger gene clusters associated with degradation of plant cell wall materials were also found. These included an 8 kb gene cluster coding for multiple pectate lyases and pectinesterases (B2R_31553-31558), and an 8 kb gene cluster encoding a family 43 glycosyl hydrolase, a pectate lyase L, a rhamnogalacturonan acetylesterase RhgT, a GDSL-like lipase/acylhydrolase, a family 53 glycosyl hydrolase, and an endoglucanase A (B2R_15915-15920). Together, these findings suggest that ASV5 *Streptomyces* has the capacity to effectively process complex organic materials shed by plant roots during growth. These results are in line with a recent study on plant-associated streptomycetes that indicated that their colonization success appears to be associated with the ability to utilize complex organic material of plant roots[54].

Root exudates also play a key role in the recruitment of microbes. Prominent sugar components of tomato root exudates are glucose, but also xylose and fructose[55]. The *Streptomyces* MAG contains *xylA* and *xylB* genes (B2R_19014, B2R_19013) and a putative *xylFGH* import system (B2R_29274, B2R_23438, B2R_23439) facilitating xylose metabolism. Similarly, a *frcBCA* import system was identified in the genome (B2R_17966-B2R_17968) as well as a glucose permease (B2R_32780) with 91,5% amino acid identity to *glcP1* SCO5578 of *Streptomyces coelicolor* A3(2)[56]. Other genes putatively involved in root exudate catabolism were also found in the ASV5 MAG, such as sarcosine oxidase (*soxBAG*, B2R_20550-20551, and B2R_21105), which has been shown to be upregulated in the presence of root exudates of various plants[57,58].

In summary, the *Cellvibrio* and *Streptomyces* contigs encoded a range of functions that likely allow them to profit from tomato root exudates as well as complex organic material shed from growing tomato roots. How these plant traits differ between wild and domesticated tomatoes and if/how these influence differential colonization of roots of wild and domesticated tomato lines by these two bacterial lineages will require detailed comparative metabolomic analyses of the root exudates of both tomato lines as well as isolation of the corresponding *Cellvibrio* and *Streptomyces* ASVs, analysis of their substrate utilization spectrum followed by site-directed mutagenesis of the candidate genes, root colonization assays and in situ localization studies.

**Genomic structure in *Cellvibrio* and *Streptomyces* provides insights into adaptations for differential recruitment.** Bacterial populations often contain significant genomic heterogeneity. This heterogeneity may be associated with differential recruitment through altered nutrient preferences or host colonization mechanisms. The use of metagenomics enabled us to investigate the population structure within each rhizobacterial lineage and identify intraspecific differences. To do so, we first identified a unique set of 697,731 microbiome Single Nucleotide Variants (SNVs) in a subset of parental and bulk metagenomes using InStrain[22]. A set of 15,026 SNVs enriched in either the wild or modern tomato rhizosphere were selected and the abundance of each allele at each SNV was calculated. Using these abundances, QTL mapping was performed using R/qtl2 as described in the methods. A total of 3,357 QTL peaks were identified (LOD > 3.01, $P < 0.05$), to 1229 independent loci. A total of 1354 QTL with positive effects and 2,001 QTL with negative effects were identified, derived from 2,898 unique SNVs, and corresponding to 810

and 1068 unique rhizobacterial genes respectively (Supplementary Data 17).

We investigated the 103 *Streptomyces* SNV QTLs at 94 unique positions within annotated genes whose mapping coincided with the previously identified QTLs for *Streptomyces* contigs to tomato chromosomes 4, 6, and 11 (Supplementary Data 17). Numerous *Streptomyces* SNVs were associated positively with the reference tomato alleles on chromosomes 6 and 11. In particular, alpha-galactosidase (B2R_16136) and arabinose import (B2R_29105) had the highest LOD and smallest overlapping confidence intervals with chromosomes 6 and 11 (Fig. 7). Indeed, many SNVs in genes involved in the degradation of xylan[59], one of the most dominant non-cellulosic polysaccharides in plant cell-walls[60], as well as carbohydrate and protein metabolism were associated positively to QTL on chromosomes 6 and 11, including xyloglucanase Xgh74A (B2R_10589), alpha-xylosidase (B2R_23763), endo-1,4-beta-xylanase (B2R_20609), extracellular exo-alpha-L-arabinofuranosidase (B2R_20608), multiple protease HtpX (B2R_19218), cutinase (B2R_19356), and putative ABC transporter substrate-binding protein YesO (B2R_09821) which has been implicated in the transport of plant cell wall pectin-derived oligosaccharides[61]. A *Streptomyces* SNV in acetolactate synthase (B2R_28001) was associated positively to QTL on tomato chromosome 6 where a plant acetolactate synthase was located. Similarly, multiple SNVs in *Streptomyces* genes involved in putrescine transportation (B2R_25489) were associated positively to QTL on tomato chromosomes 6 and 11, which contain genes for spermine synthase, suggesting a possible metabolic cross-feeding from plant to microbe. A majority of these SNVs were synonymous having no effect on the produced amino acid sequence. However, some were non-synonymous, resulting in an altered amino acid sequence, including the histidine decarboxylase SNV (B2R_16511) mapping to both tomato chromosomes 6 and 11 (Fig. 7). *Streptomyces* SNVs that were associated negatively with the QTL on tomato chromosome 4 included an antibiotic resistance gene (daunorubicin/doxorubicin, B2R_28992) and maltooligosyl trehalose synthase (B2R_07820) among others.

Similarly, we investigated the 324 *Cellvibrio* SNV QTLs within annotated genes whose mapping coincided with the previously identified *Cellvibrio* contig QTLs to chromosomes 1 and 9. Again, numerous SNV QTLs were identified in genes were related to sugar catabolism, including a gene encoding an extracellular exo-alpha-(1->5)-L-arabinofuranosidase (B2R_16093), fructose import FruK (B2R_22268), a cellulase/esterase-encoding *celE* homolog (B2R_11067), and genes involved in malate (B2R_18213), mannonate (B2R_14081), xyloglucan (B2R_10668) and xylulose (B2R_22179) metabolism. Furthermore, many additional SNV QTL were identified in genes related to vitamin and cofactor metabolism as well as sulfur and iron metabolism. In particular, these included genes for a phosphoadenosine phosphosulfate reductase (B2R_15720), vitamin B12 transporter BtuB (10 different genes, see Supplementary Data 17), a siroheme synthase (B2R_24033), a pyridoxal phosphate homeostasis protein (B2R_17481), a heme chaperone HemW (B2R_12751), a hemin transport system permease protein HmuU (B2R_09175), a Fe(2+) transporter FeoB (B2R_19968), a biotin synthase (B2R_30007), a catecholate siderophore receptor Fiu (B2R_17486), and a Fe(3+) dicitrate transport ATP-binding protein Fec (B2R_09176) (Supplementary Data 17). Taken together, this analysis suggests that a shotgun metagenomic approach integrated with quantitative plant genetics can be instrumental in a high-throughput manner to discover putative reciprocal genetic links between plant and microbial metabolisms, such as those identified here for polysaccharides, trehalose, iron, vitamin, amino acid, and polyamine metabolism.

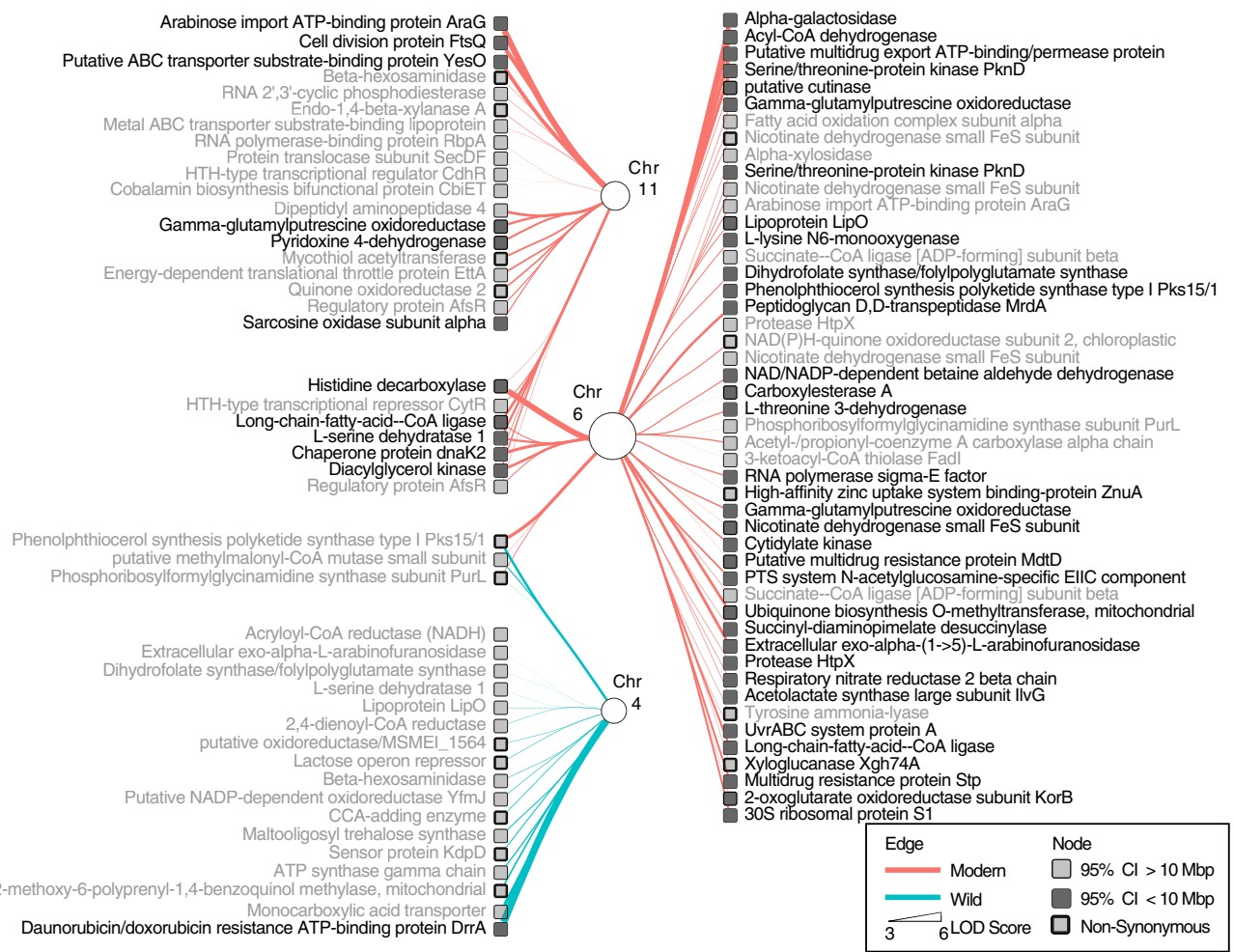

**Fig. 7 The SNP QTLs identified in the *Streptomyces* contigs mapping to the previously identified positions on chromosomes 4, 6, and 11.** The figure depicts various features of both the QTL analysis and the SNP. In particular, the edge sizes are relative to the LOD score, and edge color is coded by modern and wild. SNPs are represented by square nodes. Those with confidence intervals <10 Mbp are shaded in dark. Non-synonymous SNPs have a thick border edge. Annotations are provided next to the genes. Source data are provided as a Source Data file.

## Discussion

Breeding for microbiome-assisted crops is a daunting task, encompassing ecological, evolutionary, and cultural processes. What constitutes a desirable trait for selection is context-dependent and differs between societies, crops, and locations[62]. As society grapples with modern challenges such as a rapidly changing environment, water scarcity and land degradation, it is becoming increasingly clear that a new era of trait selection is needed with increased focus on sustainability and microbiome interactions[63–66]. In this regard, it is also time to reckon with the consequences of historic yield-centric trait selection and accompanying genomic sweeps[34], especially with regards to plant–microbe interactions (Fig. 8a, b). Current approaches to investigating the genomic architecture determining microbiome assembly rely primarily on mutational studies in known genes and pathways. More recently, studies leveraging the natural variation within plant populations have been used to conduct GWA and QTL of the leaf[20,67] and rhizosphere[18]. To date, the microbiome has been primarily characterized through amplicon sequencing, thereby providing limited functional resolution of microbiome structure. Increasing the resolution of phenotyping of quantitative traits has been shown to improve the precision and detection of QTLs[68]. Thus, integrating microbial genomics into microbiome QTL analysis plays a dual purpose; increasing the ecological resolution with which microbial traits may be

mapped (e.g., at a community and population level, Fig. 8c), and second, affording the identification of the reciprocal microbial adaptations that drive plant–microbe interactions (e.g., by using SNVs a microbiome features). In this investigation, we addressed these challenges by integrating amplicon and shotgun metagenome sequencing to identify microbiome QTLs for the tomato rhizosphere.

One major difference between the amplicon and contig QTL analysis is the number of lineages for which QTLs were identified. Amplicon-based sequencing, which captures more rare taxa per unit sequencing, provided a broader taxonomic picture and was able to capture QTLs of both abundant and relatively rare rhizobacterial lineages. In contrast, the majority of contig QTLs mapped to the most predominant lineages yet failed to identify QTLs for more rare lineages. Nevertheless, besides the fact that the shotgun-based approach provided functional insights into the associated bacterial taxa, the size of the 95% confidence interval of the QTL region was significantly smaller using contig QTLs, with a median size of just 6.47 Mbp compared to 58.56 Mbp for the amplicon-based QTL regions. Furthermore, for *Streptomyces*, the number of unique QTLs identified was greater in the contig-based approach. Thus, we identified a trade-off between amplicon and shotgun-based technologies, whereby amplicon sequencing provides a deeper view into broad community structure, whereas shotgun-based approaches provided a more nuanced picture. In

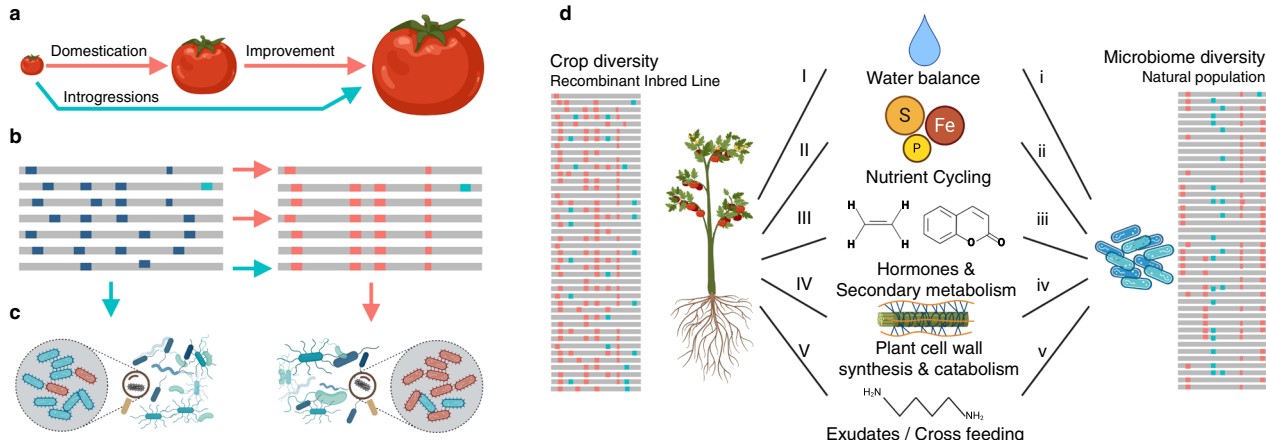

**Fig. 8 Disentangling the genetic basis of rhizosphere microbiome assembly. a** The initial domestication, subsequent crop improvements, and introgression wild tomato traits to modern cultivars. **b** While domestication significantly decreased the allelic diversity of modern tomato cultivars, introgressions of allelic diversity from wild relatives has left a genomic signature. **c** Here we identify QTLs associated with changes in microbiome composition at both the community level, but also within individual populations (e.g., *Streptomyces* and *Cellvibrio*). We show that these QTL overlapping previously identified selective sweeps associated with domestication. **d** By identifying plant QTLs regions using population features of the microbiome (SNVs), it is possible to identify the reciprocal functional adaptations that may link plant and microbe (represented by capital and lower-case letters respectively). These functions may interact directly, or indirectly via the environment. For example, related to water balance (I, i), we identified plant aquaporin and both plant/microbe trehalose metabolism. Selection for altered host water usage may alter the water balance in the soil and associated repercussions on microbiome structure. Similarly, numerous plant and microbial genes related to nutrient cycling (II, ii) involving iron, sulfur, vitamin, and phosphorus acquisition were identified. Plant signaling and hormone genes (III) identified in QTL regions included 1-aminocyclopropane-1-carboxylic acid oxidase, alpha-humulene/beta-caryophyllene synthase, and a p450 involved to coumarin synthesis. Furthermore, plant cell wall metabolism (IV, iv) including expansins, extensins, pectinesterase were linked to microbial genes involved in plant cell wall plant polysaccharides catabolism, cellobiohydrolase glycosyl hydrolases, xylose, sarcosine oxidase, L-arabinofuranosidase, fructose import, a cellulase/esterase, and xyloglucan metabolism. Finally, genes related to exudation and possible cross feeding (V, v) included plant genes such as aluminum-activated malate transporter, polyamine, glutamine, and acetolactate synthetase, and microbial functions related to malate, mannonate, polyamine, and acetolactate metabolism. Created with BioRender.com.

particular, the smaller regions identified by our contig-based metagenome mapping provided considerably more functional insights as it enabled us to analyze the genomic content contained in the regions linked to *Cellvibrio* and *Streptomyces*. It is possible that less stringent prioritization steps could be used to increase the number of metagenomic features identified, but this may also increase the false discovery rate. It should be noted that a limitation of the approaches taken is that both amplicon and shotgun-based approaches produce non-independent measurements. Here we use CSS normalization, one of the top performing computational approaches to address compositional bias[69]. Nevertheless, future approaches that provide community level absolute ASV abundances will further minimize compositionality of the microbiome data and likely perform better when mapping microbiome features as QTLs. Extending these studies to the endophytic compartment and including metatranscriptome analyses may also further improve the identification of microbiome features, provided that the endophytic microbiome can be separated well from the plant cells to obtain sufficient sequencing depth.

The increased QTL mapping resolution provided by shotgun-based phenotyping of the microbiome combined with SNV analysis provided an approach to leverage both the host diversity of the RIL and the natural microbiome population diversity to disentangle the reciprocal genomic adaptions between plants and natural microbiomes (Fig. 8d). For example, understanding the forces driving the abundances of rhizospheric *Streptomyces* is of increasing interest and has been linked to both iron[70] and water limitations[54]. Here, we pinpointed the genetic basis for these interactions among the short list of highly expressed root-specific tomato genes linked positively to *Streptomyces* abundance including both aquaporin and FIT. More specifically, the aquaporin (SlTIP2.3) has the highest fold change of all tonoplast

intrinsic proteins in the tomato genome in the roots when compared to all other organs[71,72], while the FIT gene has been shown to largely control iron homeostasis in tomato[35,73]. Future experiments will focus on functional validation by, among others, transcriptome analyses and site-directed mutagenesis of the microbial and plant genes identified.

In addition to these high priority genes, many other key genes were identified in these regions. Those previously shown to contribute to microbiome assembly included 1-aminocyclopropane-1-carboxylate oxidase, which plays a central role in plant regulation of various processes including bacterial colonization and root elongation[74] and alpha-humulene/(-)-(E)-beta-caryophyllene synthase, a terpene known to modify microbiome structure[39]. In addition, numerous genes related to growth, development, and cell wall loosening[75] known to be involved in microbial colonization[76] and aluminum-activated malate transporter, which has been linked to microbiome-mediated abiotic stress tolerance[40] and selected during tomato domestication resulting in high malate content in both fruit and roots[41]. Both low-malate and high-malate haplotypes have been identified in tomato[41], which may form the basis of future studies investigating the role of malate exudation in microbiome assembly.

The historic impact of domestication on genomic regions linked to microbiome assembly is also apparent (Fig. 6, Supplementary Data 14, and Supplementary Fig. 4). However, the processes and consequences of these sweeps, and possible subsequent recombination events on microbiome assembly remain unclear. In particular, the discontinuity of sweeps in microbiome QTL regions suggests that evolutionary pressure for recombination of key (microbiome associated) traits, such as iron homeostasis and water transport, may have acted against selective sweeps. The approach developed here provides the means to illuminate such complex eco-evolutionary questions, forming the

basis of integrating the microbiome into the classic genotype by environment model of host phenotype[10].

From the microbial perspective, the increased resolution in QTL analysis afforded by our shotgun-based approach also provided a window into the host-specific bacterial adaptations to wild and modern alleles. In particular, the SNV QTL analysis demonstrated that genes related to the degradation of various plant-associated polysaccharides in *Streptomyces* were associated positively with the modern reference allele. Many other functions were identified in both plant and microbe, such as trehalose metabolism, polyamine metabolism, and acetolactate synthase, suggesting either a direct link through cross-feeding[77] or signaling[78], or perhaps shared ecological pressures. While the microbial adaptations related to polysaccharides[79], vitamins[80] and iron metabolism[47,70] are well documented in relation to plant colonization, here we demonstrate that the reciprocal adaptations that drive plant–microbe interactions can be investigated simultaneously to uncover their genetic architecture in both host and microbiome (Fig. 8d). From a societal context, linking quantitative genetics with community level microbiome data provides us a tool to understand the complex genotype, environment, microbiome, and management interactions that shape our agroecosystems structure and function. Armed with these tools and molecular insights, we can begin to re-envision the agroecosystem; targeting QTLs for improved plant–microbe interactions, identifying 'missing microbes' or functions lost during the domestication process, or pinpointing the molecules that drive these interactions.

## Methods

**Recombinant inbred line population**. An F8 RIL population derived from the parental lines *Solanum lycopersicum* cv. Moneymaker (modern) and *Solanum pimpinellifolium* L. accession CGN14498 (wild) consisting of 100 lines were used for this study[23]. A high density map produced from this population was used to map QTLs[26].

**Growth conditions for RIL**. The natural soil was collected in June 2017 from a tomato greenhouse in South-Holland, The Netherlands (51°57'47"N 4°12'16"E). The soil was sieved, air dried, and stored at room temperature until use in 2019. Before the beginning of the experiment, soil moisture was adjusted to 20% water by volume using deionized water. All soil was homogenized by thorough mixing and allowed to sit, covered by a breathable cloth, in the greenhouse for one week prior to potting. The soil was then homogenized once again and then potted. Each pot was weighed to ensure all pots were 175 g ± 0.5 (wet weight). Duplicate pots for each accession were planted, as well as six replicates of each modern and wild parental accession, and 8 bulk soil pots that were left unseeded. Each replicate was prepared simultaneously. Planting was done separately representing biological replicates.

In each pot, 3 seeds were planted in a triangular pattern to ensure the germination success for all pots. The first seedling to emerge in each pot was retained and others were removed after germination. All pots were randomly distributed in trays containing approximately 10 plants. Throughout growth, careful attention was given to randomize the distribution of plants. First, tray location and orientation with relation to each other were randomized on a nearly daily basis. In addition, the distribution of plants within trays was randomized three times during growth. All pots were kept covered with a transparent lid until germination, which was scored daily. After germination, plants were visually monitored and watered at the same rates. To minimize the impact of environmental differences between pots on microbiome composition, the watering regime for all plants was standardized and leaks from the bottom of the pot and overflows were completely prevented. To achieve this, a minimal volume (2.5–5.0 mL) of water was used at each watering. This strategy was successful as washout was never observed. Moisture content was measured by weighing the pots at the middle and end of the experiment to ensure all pots had similar moisture contents.

**Harvesting and processing of plant materials**. All plants had between 5 and 7 true leaves at harvest (Supplementary Data 1). Plants were gently removed from the pot and roots and were vigorously shaken. Soil that remained attached to the roots after this stage was considered the rhizosphere. The remaining bulk soil and rhizosphere (plus roots) fractions were weighed. The root and attached rhizosphere fraction were treated with 4 mL of lifeguard, vortexed, and sonicated. Roots were

then removed. The remaining rhizosphere sample was then stored in LifeGuard Soil Preservation Solution (Qiagen) at −20 °C until DNA extraction.

The dry weight of shoots was measured after drying at 60 °C. The dry weight of the bulk soil was measured after storing at room temperature in open paper bags for 1 month. The DNA was extracted using the DNeasy PowerSoil extraction kit (Qiagen). The protocol was optimized for the soil in the following manner: each sample was vortexed and then a volume of approximately 1.5 mL was transferred into 2 mL tubes. This subsample was centrifuged at 10,000 × g for 30 s such that a pellet was formed. The supernatant was removed, and a new subsample was transferred, and centrifuged until the total volume of the original sample, without sand, had been transferred to the 2 mL tubes. The resulting pellet was recalcitrant to disruption through bead beating, and therefore was physically disrupted by a pipette tip before proceeding with DNA extraction protocol. In test samples, DNA extractions from the sand fraction yielding no, or marginal levels of DNA.

**rRNA amplicon sequence processing**. All DNA was sent to BaseClear (Leiden, The Netherlands) for 16S rRNA gene 300 bp paired-end amplicon sequencing (MiSeq platform). MiSeq primers targeted the V3-V4 region of Bacteria: 341FCCTACGGGNGGCWGCAG, 805RGACTACHVGGGTATCTAATCC. In total, 20,542,135 16S rRNA gene amplicon read pairs over 225 samples were generated. The raw reads were processed using the DADA2 workflow (v1.14.1) to produce amplicon sequence variants (ASV) and to assign taxonomy based on the Silva database version 138[81,82] (Supplementary Data 2). ASVs tagged as non-bacterial, chloroplast, or mitochondria were removed. Next, ASV counts were normalized using the cumulative sum scaling (CSS) (Supplementary Data 3), which has been shown to be one of the most effective computational transformation techniques[69], and filtered based on the effective sample size using the metagenomeSeq package (v1.28.2)[27]. Differential abundances between rhizosphere and bulk soil were determined using the eBayes function from the limma package. Enriched rhizosphere ASVs with a greater than log(2) fold change in abundance were analyzed based on their presence and absence, standard deviation and mean values. Using these statistics, stochastic ASVs (<50% of samples) were removed from further analysis (Supplementary Data 4). All ASV sequences may be found in Supplementary Data 5. The remaining microbiome features were then mapped as QTLs as described subsequently. To investigate diversity metrics as quantitative traits, the Shannon diversity of each sample was calculated using all ASV after filtering based on the effective sample size using the metagenomeSeq package (v1.28.2)[27], and using all ASV in greater than 50% of samples (Supplementary Data 21). Similarly, a PCoA analysis using Bray Curtis distances was conducted, and the values for principle components axis 1 and 2 were extracted (Supplementary Data 22). Both calculations were done in phyloseq version 1.34.0[83]. These diversity-based microbiome features were then mapped as QTLs as described subsequently.

**Metagenomics analysis**. For the one set of replicates for each accession, paired-end sequence read libraries were generated in the length of 150 bp per read on NovaSeq paired-end platform by BaseClear B.V. Demultiplexing was performed before the following analysis. It is computationally expensive to assemble the 114 read libraries all at once. Therefore, a strategy of (merging) partial assemblies was undertaken. Two assemblers were used to create the assembled contigs, namely SPAdes (version 3.13.2)[84] and MEGAHIT (version 1.2.9)[85]. Assembly quality was assessed by running MultiQC (version 1.8)[86] with Quast Module[87] (Supplementary Figure 5). First, 6 modern parents, 5 wild parents, and 1 bulk soil sample were co-assembled via SPAdes with the metagenomic mode and parameter of -k 21,33,55,99, generating the first assembly (A1). Subsequently, a second assembly (A2) was done using the unmapped reads from the remaining metagenomes using MEGAHIT with the parameter of --k-list 27,33,55,77,99. The third assembly (A3) was performed similarly as A2, however, included the unmapped reads, ambiguously mapped reads, and mapped reads with a low mapping quality score (MapQ < 20) (Supplementary Data 18). Read mapping was done with BWA-MEM with default settings[88] and SAMtools was used to convert the resulting SAM files into sorted and indexed BAM files (version 1.10). Extraction of these reads was conducted by samtools bam2fq. Redundancy between assemblies was evaluated by alignment to A1 via nucmer package of MUMmer with --maxmatch option (version:4.0.0)[89].

Firstly, 111.5 Gbp of reads from the parental samples were assembled, labeled as A1, and yielded a total assembly length of 8.6 Gbp with the largest contig of 933.0 kilobase pairs (Kbp). After aligning the reads from RIL samples to A1, unmapped reads, ambiguously mapped reads, and mapped reads with a low mapping quality score (MapQ < 20) were retrieved and assembled, yielding the second and third assembly (A2 and A3). Specifically, A2 stemmed from solely the unmapped reads while A3 included the ambiguously mapped reads and mapped reads with MapQ < 20 in addition to the unmapped reads. A2 and A3 produced a total assembly length of 9.6 Gbp and 14.0 Gbp, with the largest contig of 56.2 and 86.3 Kbp respectively. There were 1.2, 2.0, and 2.8 million contigs with the length over 1 Kb for A1, A2, and A3 respectively. In particular, 912 contigs in A1 were greater or equal to 50 Kbp whereas 1 or 2 such large contigs were successfully assembled in A2 or A3. The detailed assembly statistics is given in Supplementary Data 18 and the numbers of contigs with different ranges of length for each assembly are presented in Supplementary Fig. 5.

The sequence similarities of the contigs in each assembly (≥1 Kbp) were compared using the nucmer package in MUMer. No contigs in A2 were reported to share an overlapped region with A1, therefore contigs in A1 and A2 could be merged directly. When A3 was aligned to A1, 1.1% of the total length (≥1 Kbp) of A3 was reported to be overlapped with A1, however, only 18 contigs from A3 were 100% identical to regions in larger contig in A1. The sensitivity of filtering the overlapping contigs was evaluated by a benchmarking test using a random RIL sample to calculate the mapping rates (Supplementary Fig. 6). 83.4% reads were mapped to A1 + A3 at MapQ ≥ 20 without filtering. Excluding the contigs from A3 that were completely and identically covered by A1, the mapping rate was nearly the same as the one without filtering. Nevertheless, the removal of all aligned contigs in A3 resulted in a slight drop of mapping rate to 82.6%. To conclude, the final assembly was determined as A1 + A3 with the 18 redundant contigs from A3 removed.

To assess the overall assembly quality and quantify the abundance of contigs among all samples, metagenomic reads were mapped to A1, A1 + A2, and A1 + A3 (deduplicated) respectively. Afterwards, the mapping rates were calculated for the mapped reads with MapQ > 20 in each sample. As shown in Supplementary Fig. 7, approximately 70% reads among rhizosphere samples could be mapped to A1, while the mapping rates were 55 to 65% in the bulk soil samples. With the unmapped reads assembled and added to A1, the mapping rates for A1 + A2 increased by 10%. The read recruitment was further improved by assembling and adding ambiguously mapped reads and mapped reads with low MapQ in the final assembly (A1 + A3). A1, as well as de-replicated A3, were merged to acquire the final assembly. All the 'contigs' mentioned below are referring to the contigs in this final assembly.

**Binning of metagenomic contigs.** Metabat2 (version 2:2.15)[90] was used for assigning the contigs into genomic bins. Based on tetra-nucleotide frequency and abundance scores, 588 genomic bins were generated. Afterwards, genomic quality of those genomes was evaluated by CheckM (version: 1.1.1)[30] with the command "checkm lineage_wf" (Supplementary Data 8). The 33 genomes displaying the completeness larger than 90% and contamination smaller than 5% were used for further study as quantitative traits.

**Making phenotype files based on contig depth.** Read counts for each position on the assembled contigs were acquired using bedtools genomecov (version: 2.29.2)[91]. A custom Python script was applied to calculate the average depth (defined as the number of total mapped reads divided by contig length) and coverage (defined as the number of covered base pairs divided by contig length) of every contig. Furthermore, the average abundance of contigs assigned into a bin was calculated for the high-quality genomic bins detected by CheckM[30].

**Feature selection.** Average depths of the contigs were first normalized using the CSS and filtered based on the effective sample size using metagenomeSeq package (v1.28.2)[27]. Differential abundance analysis was performed by moderated t-tests between groups using the makeContrasts and eBayes commands retrieved from the R package Limma (v.3.22.7)[92]. Obtained P-values were adjusted using the Benjamini–Hochberg correction method. Differences in the abundance of contigs between groups were considered significant when adjusted P-values were lower than 0.01 (Supplementary Data 19).

In either comparison, the contigs that were significantly enriched in the rhizosphere were gathered and regarded as the statistically rhizosphere-enriched contigs after removing the replicated ones. To perform QTL analysis for the abundance of these enriched rhizosphere contigs, only the contigs with biological meanings were kept, i.e., the log (2) fold-change of mean values for the normalized abundances of RIL and bulk samples should be greater than 2, and the contig should be in enough depth with at least the mean value of a group larger than 1. This selection step resulted in 1249 rhizosphere-enriched contigs. The statistics of the filtered normalized abundance were further inspected based on the presence and absence of contigs, standard deviation, and mean values of the counts.

**Taxonomic and functional annotation of the metagenome.** Taxonomic classifications were assigned to the contigs in the final assembly using Kraken2 (version: 2.0.8)[31] based on exact k-mer matches. A custom Kraken2 database was built to contain RefSeq complete genomes/proteins of archaea, bacteria, viral, fungi, and protozoa. Univec_Core was also included in the custom database (20200308). Using the Kraken2 standard output, a python script based on TaxonKit[93] was utilized to add full taxonomic names to each contig in the format of tab-delimited table. 76.22% of the contigs > 1 kb were classified. Among the contigs >10 kb, up to 99.44% contigs were classified. Prokaryotic microbial genes were predicted by Prodigal (version: 2.6.3)[94] with metagenomics mode. 10,246,55 genes were predicted from contigs > 1 kb. Open reading frames (ORFs) on contigs >10 kb were annotated by prokka (v1.14.5) and the *Streptomyces* ASV5 bin (MAG.72) was further annotated by DRAM (v1.2.0) integrating UniRef, Pfam, dbCAN and KEGG databases[95]. To assess the impact of the prioritization on the functional representation of the metagenome, we identified the fraction of protein clusters represented in the rhizosphere-enriched contigs compared to the rest of the contigs greater than 10 kb. First, Prodigal was used in metagenomics mode to predict genes

in the metagenomic assembly with contigs longer than 10 kbp. Next, MMSeqs2 was used to cluster the protein sequences based on 70% similarity and based on 50% similarity, and with or without partial predicted genes[32]. To calculate the number of clusters that contained proteins encoded in rhizosphere-enriched contigs, the clusters were searched for the presence of protein IDs of the 1249 rhizosphere-enriched contigs. In total, approximately 8.3% of protein clusters contained genes from the rhizosphere-enriched contigs. In addition to proteins contained on rhizosphere-enriched contigs, these clusters contained approximately 25% of all proteins encoded in contigs larger than 10 kb (Supplementary Data 20).

**Single nucleotide variant analysis.** To investigate strain level QTLs, we mapped single nucleotide variants (SNVs) identified using inStrain on the 1249 rhizosphere-enriched contigs. A total of 555, 382, and 535,432 SNVs were identified in the modern and wild parental metagenomes respectively. Of these, 162,299 and 142,349 SNVs were unique to each dataset respectively, as they either contained only reference alleles or did not exceed the inStrain SNV calling thresholds. For each unique SNV locus, coverage in the other dataset was determined using SAMtools depth after read filtering with settings comparable to inStrain and was considered identical to the reference allele frequency. Including the unique SNVs, this resulted in a final set of 697,731 SNVs. To select SNVs that showed differential reference allele frequencies between MM and P, first the difference in reference allele frequency (MM–P) was calculated per SNV. From the distribution of all SNVs, the 95% confidence interval (CI) was determined to select the 5% (30,911) most different SNVs (Supplementary Fig. 8). SNVs were further selected using a Fisher's exact test based on the allele read count differences between MM and P. P-values were sorted, and a final selection of 15,026 differentially abundant SNVs distributed over 1037 contigs was obtained using a Benjamini-Hochberg false discovery rate (FDR) correction of 0.01. SNV allele read counts were extracted from the RIL dataset using the pysam Python package after filtering with settings comparable to inStrain.

**Quantitative trait locus analysis.** The QTL analysis linking selected amplicon, contig, bin, and SNV features with plant loci was performed using the R package R/qtl2[25]. Pseudomarkers were added to the genetic map to increase resolution, with a step distance of 1 Mbp between the markers and pseudomarkers. Plant genome probabilities were calculated using the genetic map with pseudomarkers, plant loci cross data, and error probability of 1E-4. Plant locus kinship matrix was calculated as proportion of shared alleles using conditional genome probabilities of all plant chromosomes, which were calculated from the plant genome probabilities. A genome scan using a single-QTL model using a linear mixed model was performed on the SNV allele read counts as phenotypes, plant genotype probabilities as input variables and as covariates the number of leaves, harvest day, rhizosphere soil weight (g), soil starting weight (g) and plant dry weight (g). The LOD score was determined for each plant locus SNV combination. A permutation test using randomized data was performed with 1000 permutations to assess the distribution of the LOD scores. The 95% quantile was used as threshold for the selection of LOD peaks, as well as a P = 0.95 Bayes credible interval probability.

**Independent validation of QTLs through bulk segregant analysis.** To validate the QTLs, 33 *Solanum lycopersicum* cv. Moneymaker (modern), 30 *Solanum pimpinellifolium* L. accession CGN14498, and 77 RIL accessions (with replicates of 4 each) were grown and their microbiomes characterized through 16S rRNA gene amplicon sequencing. Parental lines and RIL accessions were germinated in pots filled with 300 g agricultural soil. For each accession, were planted with six plants per replicate pot. The plants were arranged randomly in the growth chamber (25 °C, 16 h daylight) and watered every day. Bulk soil samples without plants were used as controls (N = 31).

Rhizospheric soil was collected according to standard methods[96]. In order to synchronize the developmental stage, the plants were harvested after 21 days, or when the 3rd trifoliate leaf was reached. The soil loosely attached to the roots was removed and the entire root system was transferred to a 15 mL tube containing 5 mL LifeGuard Soil Preservation Solution (MoBio Laboratories). The tubes were vigorously vortexed and sonicated. Subsequently, the roots were removed and at least 1 g (wet weight) of rhizospheric soil was recovered per sample for DNA extraction. For the bulk soil samples, approximately 1 g of soil was collected and mixed with 5 mL of LifeGuard solution.

To extract rhizospheric DNA, PowerSoil Total DNA/RNA Isolation Kit (MoBio Laboratories, Inc., USA) was used in accordance with the manufacturer's instruction. Rhizospheric DNA was obtained using RNA PoweSoil DNA Elution Accessory Kit (MoBio Laboratories, Inc. USA). The quantity and quality of the obtained DNA was checked by ND1000 spectrophotometer (NanoDrop Technologies, Wilmington, DE, USA) and Qubit 2.0 fluorometer (ThermoFisher Scientific, USA). DNA samples were stored at −20 °C until further use.

The extracted samples were used for amplification and sequencing of the 16S rRNA gene, targeting the variable V3–V4 (Forward Primer: 5′-CCTACGGGNG GCWGCAG-3′ Reverse Primer: 5′-GACTACHVGGGTATACTAATCC-3′) resulting in amplicons of approximately ~460 bp. Dual indices and Illumina sequencing adapters using the Nextera XT Index Kit were attached to the V3–V4 amplicons. Subsequently, library quantification, normalization, and pooling were

performed and MiSeq v3 reagent kits were used to finally load the samples for MiSeq sequencing. For more info please refer to the guidelines of Illumina MiSeq System. The RDP extension to PANDASeq[97], named Assembler[98], was used to merge paired-end reads with a minimum overlap of 10 bp and at least a Phred score of 25. Primer sequences were removed from the per sample FASTQ files using Flexbar version 2.5[99]. Reads were processed as before except the Silva version 132 was used for taxonomic classification[82].

**Reporting summary**. Further information on research design is available in the Nature Research Reporting Summary linked to this article.

## Data availability

The 16S amplicons and shotgun metagenomics sequencing data have been deposited in the NCBI database under BioProject ID PRJNA787039 and PRJNA789467, respectively. Metagenome assembled genomes are available at Zenodo [https://doi.org/10.5281/zenodo.6561541]. The Silva database was used to assign taxonomy to 16S rRNA amplicon sequences [https://www.arb-silva.de/download/archive/]. A custom database was used to assign taxonomy for Kraken. Due to size limitation, this database is available upon request (please contact J.M.R. at j.raaijmakers@nioo.knaw.nl and expect 2 weeks of processing time). Source data are provided with this paper.

## Code availability

The code used in the analysis can be found at Zenodo [https://doi.org/10.5281/zenodo.6561541].

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

## Acknowledgements

The project was financially supported, in part, by the NWO-TTW Perspective program BackToRoots (TTW-project 14218 to J.M.R., V.J.C., V.C., and B.O.O.), by the NWO-Gravitation program MICRop (to J.M.R., M.H.M.), a National Institutes of Health (NIH) Genome to Natural Products Network supplementary award (no. U01GM110706 to M.H.M.), a ZonMW Enabling Technologies Hotel project (no. 40-43500-98-210 to M.H.M.), a Senescyt fellowship awarded to S.S.F., and by internal funding from the Netherlands Institute of Ecology.

## Author contributions

The study was conceived and designed by B.O.O., V.J.C., W.Li, M.H.M., and J.M.R. The greenhouse experimentation and lab work were conducted by B.O.O., S.S.F., V.C., V.J.C.,

and A.N. Contributions to data analysis came from B.O.O., T.G., X.P., E.v.d.W., W.Lo, L.P., N.S., A.K., V.C., V.J.C., B.L.S., M.H.M., J.N.P., and M.M. The manuscript was drafted by B.O.O., B.L.S., M.H.M., and J.M.R. All authors contributed to the revision and agreed upon the final draft.

## Competing interests

The authors declare no competing interests.
