## [Peer Review File · Nature Communications]

Disentangling the genetic basis of rhizosphere microbiome assembly in tomatoReviewers' Comments:

Reviewer #1:

Remarks to the Author:

The manuscript by Oyserman et al. uses a high-resolution mapping population to describe the genetic architecture of microbiome and metagenome features in tomato rhizospheres. A standard QTL mapping approach was used to study the genetic basis of three sets of microbial features: (1) the abundance of individual taxonomic groups/ASVs, (2) the abundance of metagenome fragments, and (3) allele frequencies at SNPs within bacterial metagenomes. It is rare to see these three types of data in the same project, and provides interesting insights into the overlap, and sometimes lack thereof, in the plant genes affecting these features. Analysis of the microbial genes that are associated with plant QTL provides additional insight into which bacterial genes or functions may be involved in interactions with the plant. However, the main output of this work is a list of candidate tomato genes that will still need to be functionally validated, and the concept of treating microbiome features as quantitative plant traits and applying quantitative genetics approaches is not novel. Nevertheless this ambitious manuscript is a valuable source of information about the genetic architecture of an extremely complex, but likely ecologically important "extended phenotype". I applaud the rigorous and detailed approach to these complex analyses, and am very happy to see the data and code already available, which will allow further dissection of this very rich dataset.

My main concern about this manuscript is related to the inferences that some QTL affect a relatively large number of microbiome features - for example, the hotspot identified on chromosome 11 that controls the abundance of many ASVs (Fig. 3d). Although it's certainly plausible that such hotspots could exist, because the phenotype here is measured using sequencing data, the measured abundances of separate ASVs are actually non-independent from each other: for a given number of sequence reads (a ceiling set by the output of the sequencer, unrelated to the biology of the system), if the true abundance of one ASV goes up, then the measured abundances of all other ASVs will go down. Metagenome contigs have the same problem. This well-known compositionality problem creates spurious negative correlations between features, which in turn could create the illusion of a shared genetic basis where there is none. If the counts were to be corrected for their compositional nature (see e.g., DOI: 10.1093/gigascience/giz107) would these hotspots still be detected?

I also have a somewhat major critique of the terminology used to describe the significant QTLs and especially their effect sizes. Throughout this paper, each QTL is described as either "wild" or "modern" - in reference to the two parent genotypes of the RIL population. However, in a classic quantitative genetics framework, a QTL is simply a location in a genome where there are 2 segregating alleles with differing effects on the trait of interest; therefore it cannot have the property of "belonging" to one parent or the other. In contrast, each allele at each QTL does come from one of the two parents. Nevertheless, it is not coherent to describe an ASV or other metagenome feature as being linked to either a "wild" allele or a "modern" allele (two examples of many: lines 67-68 and 119). This is because it is the difference between both alleles that defines a QTL. The overall effect of this incorrect terminology is to create confusion about the actual size and direction of the alleles' effects. In my opinion the simplest and clearest way to fix this problem would be to set one of the two parent alleles as the "reference" and then describe the QTLs not as "wild or modern" but rather "positive or negative" relative to the reference allele. For example, the effect size & direction of the modern allele could be reported for all QTLs. This would be much easier to understand biologically, e.g. it would be easy to understand that an ASV became more abundant or less abundant due to genetic changes during domestication.

Some more minor comments:

Line 52 "This approach has been adopted" implies that microbiomes are already being used as breeding targets, which as far as I know is not true. In general this paper creates a false impression that "breeding for the microbiome" is feasible at the scale needed for crop improvement - I suggest adding some caveats about the challenges of knowing which microbiome features to target for crop

improvement.

Line 113: What does it mean for a QTL to be "more abundant in" an allele?

Lines 130-132 and 134: Unclear what effect sizes are being compared

Lines 153-155: Clarify that the "rhizosphere enrichment" is relative to bulk soil data

Lines 151-155: Is there a way to estimate what proportion of the functional diversity was excluded from the metagenome dataset due to these steps? (I am not suggesting that all 40 million contigs should have been tested!)

Lines 162-163: QTLs "belonging to" bacterial taxa is confusing - the QTLs are in the plant genome and the contigs are from bacterial genomes. In general, these complex issues require very careful writing, I recommend being very explicit with the language: "QTLs underlying *Streptomyces* contigs", for example

Lines 208: Similar to above comment: clarify that "root specific genes" are plant genes, not *Streptomyces* genes that are expressed when colonizing the root. Lots of other examples of similar problems

Line 216: "Of interest" - explain what makes a candidate gene "of interest" for this analysis.

Lines 237-238: Clarify that if this association is real, the direction of causality is still unknown, the decline in *Cellvibrio* abundance could be a "side effect" of domestication rather than a mechanism of it. The same is true for the interpretation of all other QTL effects

Line 325: "*Streptomyces* contig QTLs" → "QTLs for *Streptomyces* contigs"

Lines 330-332: An example of where the writing gets confusing because it simultaneously discusses genes in bacteria and plant. For example, clarify "many SNPs" → "many bacterial SNPs" and "chromosomes 6 and 11" → "tomato chromosomes 6 and 11" as often as possible.

Lines 338-340: This is an interesting observation. In general I really like the attention paid to genetic variation within microbial lineages

Line 349 and several other places: SNV is used instead of SNP to mean the same thing, recommend consistent terminology

Lines 362-365: Reiterate here that functional validation is still required for these candidate genes.

Line 367: "Daunting task" is a better descriptor than "complex phenomenon" in my opinion, as no one is really doing this yet as far as I know

Lines 395-397: Could this discrepancy between amplicon and shotgun data be partially explained by the way most of the contigs had to be discarded, and/or the differences in the completeness of the reference databases used?

Line 409: Fold change compared to what?

Line 449: Were the tomatoes being grown in natural soil, or in commercially-provided potting mix?

Line 465: Covered with what? I assume a transparent lid to allow light?

Line 499: What reference database was used to assign taxonomy?

Lines 506+: I appreciate the high level of detail for the metagenomics analyses, but they do seem unbalanced relative to the amount of detail provided for the amplicon data.,

Line 631: What exactly was the linear mixed model - for example, which of the predictors were included as random vs. fixed effects?

Line 636: Permutations of what?

Line 652: It's unclear what the extracted RNA was used for (as opposed to the DNA used for 16S sequencing)

Reviewer #2:

Remarks to the Author:

the authors used microbiomes of an offspring population of a wild and modern tomato as an external phenotype for QTL analysis. They identified genetic regions that correlated with the association of specific microbes. With metagenome analyses, they additionally identified microbial traits associated with wild or modern tomato QTLs.

t=/the results and figures are presented in a clear, concise way, the manuscript is well written, and

the conclusions are supported by evidence. The authors utilized a broad range of modern techniques. Their methodological approach as well as their biological findings will be of interest for a broad readership.

Detailed comments:

L 96/ Fig 1: It is laudable that the authors characterize the RIL population general traits. As these results have been published before and as they do not directly impact the work presented here, I would move this figure to supplemental data.

L108/Fig2A: most of the variation is found between bulk soil and rhizosphere samples, which is consistent with other studies. Please also include a PCA plot of rhizosphere samples only to better illustrate differences between the tomato lines.

L210: please introduce the concept of selective sweeps.

L238: the authors identified quite a number of *Cellvibrio* and *Streptomyces* genes associated with the different QTLs. Are these genes specific to the bacterial strains identified here/ associated with plants or are these common features found in many related strains as well that do not interact with plants?

L341: please explain the synonymous/ nonsynonymous terminology.

best regards, Joelle Schlaepfer

Reviewer #3:

Remarks to the Author:

In this manuscript, the authors perform QTL mapping in tomato inbred lines, using the root microbiota as the quantitative trait. This innovative method enabled them to reveal new genes in both the plant and the microbiota, involved in microbiome assembly. The combination of both 16S rRNA gene amplicon and metagenomics analysis allowed the authors to map QTLs involved in microbiome assembly both broadly (16S) and relatively precisely (shotgun). This method, facilitated by a wide comparison of inbred lines of a cross between wild and commercial tomato varieties, supplies a thorough insight into tomato genetic basis for microbiome assembly.

This manuscript will be of very high interest to the scientific community and makes an interesting read. I have only minor comments, mostly regarding the presentation of the data, but I also raise some methodological issues. In some parts of the manuscript, a greater attention to detail is needed. A validation of some of the loci identified using genomic editing in both the plant and the microbiota would have been very welcome, and I hope the authors will perform them down the road, but I think that the amount of work presented is certainly enough for an impactful paper.

One important comment that I have regards the paper's take-home message. Many functions are listed, but it is not made clear if how they are all connected and if there is an emerging pattern. Perhaps a summary figure would help illustrate this.

The approach the authors take is to consider the relative abundance of each microbial taxon within the microbiota as a quantitative trait. However, this skips over the use of quantitative ecological measures of the community as a whole. Using the methods applied here it should be straightforward to identify QTLs for microbiome alpha and beta diversity (for the latter, a PCoA without bulk soil could be calculated, and then the values along the 1st and 2nd axis could be used as quantitative traits). Another measure that would potentially be very interesting to look at is absolute bacterial abundance. It is likely that there would be QTLs that correlate with the general ability of microbes to colonize the

roots. Unfortunately, this manuscript, as many others, does not consider absolute abundance. Perhaps by using the metagenomic data the authors could devise some kind of proxy (e.g the ratio between bacterial and host reads in the data).

Comments in order of appearance in the text:

Line 18: 16S amplicon is too much of a shorthand in my opinion. I suggest 16S rRNA gene amplicon

Line 39: delete "a"

Line 40: when you say "microbiome breeding programs", if I understand correctly, you mean plant breeding for specific microbiome selection, rather than breeding of the microbiota themselves. I think that the current wording could be somewhat ambiguous in this respect.

Line 55: should be "in their infancy" (I think)

Figure 1

Here you use either scatter plots or box plots. Why the inconsistency? Elsewhere you use combined box and scatter plots, which are more informative and transparent. Why not use that format for this figure as well?

The y-axis height for panels c and d and e and f are not exactly the same.

"containing neither allele (labeled zero)" this wording is confusing. It is shorthand for "neither BB on 2 and AA on 9", but they are not referred to yet at this point in the sentence.

How many traits were tested in total?

How does this QTL mapping compare in general to the gnotobiotic QTL mapping?

How do the authors interpret contrasting results?

Line 110: It would be nice to see how abundant and ubiquitous are these ASVs in the data? Perhaps a version of Fig. 2b could be made with these 33 ASVs marked by a different color, and added as a supplementary figure.

Line 117: Here it says 14 taxa out of 25 but above in line 110 you say you found 33 ASVs. What do you mean here by "taxa" and how does this square with the number of ASVs? From the taxonomic names in Figure 3 I understand that abundance at all taxonomic ranks was considered, but I could not find this explained anywhere. Perhaps I missed it, but this could be clarified better here.

Figure 3:

Panels are labeled with uppercase letters in the text and lowercase letters in the figure (here and elsewhere).

The effect size units in panel C could be more explicit. Perhaps explain this more in the legend. It is not trivial enough to just be denoted by a % sign (on a related note - why is it important to statistically compare effect sizes among taxa? The rationale for this analysis is not made clear in lines 125-140). Also, the text refers first to panel D and then to panel C. That's confusing.

The color shades of panels B and C don't match. What are the colors there for anyway?

Panel D:

You do not explain what the edge colors denote. I assume they correspond to the wild/modern colors in panel A but the reader does not have to figure that out on their own. Also the color shades don't match. In addition, some of the nodes seem to change color when passing through the edges. The figure caption ends with the statement that "A complex network emerges". This is a rather diffuse statement. Complex in relation to what? Much of the complexity of this network results from the choice to include multiple taxonomic ranks in preparing it. Moreover, many data structures can be presented in a network form, but that does not suffice to conclude that this is a truly interacting

network as implied here. All in all I get the impression that this figure panel was included because it looks cool (it really does) but that the authors struggle to draw a meaningful conclusion from it.

Line 145: looks like one sample was switched in error (no need to make any changes here, these things happen...).

Line 148: At least some readers will find this confusing. I think a sentence would be in place here to say that bin and contig abundances were determined by read depth? I know it appears in the Methods, but please spell out CSS here. It is a bit unbalanced to rush through the normalization and abundance calculations here, but on the other hand devote a sentence to list which software was used for binning. In general, I think some of the details in the Methods need to be included here, like the fact that you devised a strategy to co-assemble ALL metagenomes, allowing you to map reads from all samples to the contigs and perform a differential abundance analysis. I think that a reader that skips looking at the methods should still be made aware of this A1-A2-A3 strategy.

Line 151: Many open questions here: how do the taxonomic profiles from metagenomics and 16S compare? Do the soil, Moneymaker and wild samples separate on a PCoA space similarly as the 16S data? How many of the contigs were not bacterial? How were these used downstream?

Line 152: "we took numerous ..." this is an odd choice of words. I can't imagine you used so many prioritization steps that you couldn't count them. Can't you just say how many steps?

Figure 4: See comments for Figure 3. Also, I understand why you use a log scale here in panel B but I cannot understand why the boxes are all different sizes. Very confusing. Isn't each box one contig QTL? What is count+1,2?

Figure 5

What are the units of the y-axis? The legend says relative abundance? So why are the values so high? Why do the colors signify? Seems like they have no meaning, in which case I suggest removing them. This is doubly confusing considering a similar palette was used to signify something else in previous figures.

It does not say anywhere what M and P stand for. I assume these are the parental lines? M for Moneymaker/Modern? P for pimpinellifolium? I don't get it.

Figure 6

This figure sends the reader to do too much homework. y-axis should say $\log_2(\text{root/leaf expression ratio})$. The legend title for panel b should not be "Legend (FPKM)". You could change this to "Transcript abundance ($\log_2(\text{FPKM})$).". You do not spell-out FPKM anywhere.

Line 206: RNA-seq

Line 236: it is not entirely clear what the authors think is the relationship between these genome-wide sweeps and the microbiota are. Is it causal, or more coincidental? Could the authors try and explain this and the evidence supporting one or the other more precisely?

Lines 304-309. I completely agree. In a previous paper (Carrion et al, 2019) the same authors actually took this step and validated metagenomic hypotheses in by site directed mutagenesis. Here such validations can be done with genomic editing in both bacteria and plants, which would substantially elevate the impact of this manuscript. I am not suggesting however that such experiments should be a prerequisite for publication in my opinion.

Line 365: This manuscript unloads a huge amount of information on the reader. The authors did a good job at highlighting in the text numerous functions in both the plant and the microbiome that the

analysis identified. Some kind of synthesis is needed here, to highlight the patterns that emerge. The take-home sentence at the end of the Results section focuses on the method itself rather than the findings. I think that this is a missed opportunity. A graphical summary of the findings would go a very long way here to highlight the patterns of plant-microbe interactions that emerge from this study.

Lines 396-397: Can you provide an explanation for this tradeoff? Is it because metagenomics goes into a deeper taxonomic resolution while amplicon allows us to see less abundant taxa? That seems like the obvious answer but I think you should spell it out.

Line 405: "driving forces driving" change to "forces driving".

Methods:

This paper employs many state-of-the-art methods. I suggest that to make things easier on the reader, the authors add to each sub header which section in the Results and which Figure this section refers to.

Line 499: Supplementary table 2 still contains the mitochondrial and chloroplast reads. On that matter, the supplementary tables are provided with no caption or explanation and with meaningless file names. Please amend that. In addition, in the text, (Line 100) you refer to sup. Tables 2-5 in bulk, without explaining what each one is.

Line 516: isn't there a sentence missing here saying something along the lines of "then we mapped all reads from the 96 RIL samples to this assembly"?

Line 564: "genomic genomics"

Line 606: here and in the corresponding Results section, can you explain how SNPs and SNVs relate to one another? There are a few other things here that I find confusing. What are the "1249 contig enriched genomes"? In an earlier paragraph you say that you identified 1249 rhizosphere enriched contigs. So is this SNP analysis done on contigs or on genomes?

Or on MAGs containing these 1249 contigs? I'm confused.

Furthermore, in either case, how does SNP calling work when you are looking at a diverse collection of genomes /contigs not necessarily belonging to a single lineage? Do you only consider core orthologous genes? Do you map reads to these contigs to identify any polymorphisms among the mapping data? If so, is it the same mapping as before or do you re-map the reads? Please add some more details to explain the analysis pipeline better.

Reviewer #4:

Remarks to the Author:

The manuscript submitted by Oyserman and colleagues presents a beautiful work on investigating the genetic basis of rhizosphere microbiome assembly in tomato. Manuscript is well written and clearly structured. Authors provided solid data in both host plant and the microbiome. Main conclusions can be supported by the provided evidence. I agree that this study shed light on a new approach for further understanding plant microbiome interactions and will be surely benefit for plant microbiome breeding program.

Specific comments:

1. Except for QTLs identification, it is worth to check phenotypical variation of the RIL populations compared to modern *Solanum lycopersicum* var. MoneyMaker and wild *Solanum pimpinellifolium*. Linking genetic architecture with phenotypic observations and the microbiome assembly will make the investigations of this study more powerful and meaningful.

2. Both wild and modern alleles were identified, large number of modern alleles were identified with highly associations of rhizosphere microbiome assembly. I wonder what if this experiment was conducted under a field condition with natural soil associated with local microbiomes. In this situation, I would expect larger number of wild alleles than modern will be identified with highly association of microbiome assembly. So, did you perform further analysis to prove your conclusions with any soils which differs to the commercial greenhouse soil?

3. Rhizosphere microbiome composition is closely related to root exudates. Compared to bulk soil communities, rhizosphere reflects higher plant hosts selection that influence microbiome assemblies. Considering rhizosphere effects, compared to the rhizosphere, endophytic compartment will have even stronger plant host selection. Plant genetic variation has stronger effects on microbiome assembly in the endophytic compartment. Please discuss why in this study authors considered focusing on rhizosphere microbiome instead of microbiomes in endophytic compartment.

4. Validation of Cellvibrio and Streptomyces 16S rRNA QTLs with bulk segregant analysis showed in Fig.5. This independent experiment was conducted with modern, wild and 77 RIL accessions. Why didn't use all 96 RIL accessions? Please further clarify it.

REVIEWER COMMENTS

Reviewer #1 (Remarks to the Author):

The manuscript by Oyserman et al. uses a high-resolution mapping population to describe the genetic architecture of microbiome and metagenome features in tomato rhizospheres. A standard QTL mapping approach was used to study the genetic basis of three sets of microbial features: (1) the abundance of individual taxonomic groups/ASVs, (2) the abundance of metagenome fragments, and (3) allele frequencies at SNPs within bacterial metagenomes. It is rare to see these three types of data in the same project, and provides interesting insights into the overlap, and sometimes lack thereof, in the plant genes affecting these features. Analysis of the microbial genes that are associated with plant QTL provides additional insight into which bacterial genes or functions may be involved in interactions with the plant. However, the main output of this work is a list of candidate tomato genes that will still need to be functionally validated, and the concept of treating microbiome features as quantitative plant traits and applying quantitative genetics approaches is not novel. Nevertheless this ambitious manuscript is a valuable source of information about the genetic architecture of an extremely complex, but likely ecologically important “extended phenotype”. I applaud the rigorous and detailed approach to these complex analyses, and am very happy to see the data and code already available, which will allow further dissection of this very rich dataset.

My main concern about this manuscript is related to the inferences that some QTL affect a relatively large number of microbiome features - for example, the hotspot identified on chromosome 11 that controls the abundance of many ASVs (Fig. 3d). Although it's certainly plausible that such hotspots could exist, because the phenotype here is measured using sequencing data, the measured abundances of separate ASVs are actually non-independent from each other: for a given number of sequence reads (a ceiling set by the output of the sequencer, unrelated to the biology of the system), if the true abundance of one ASV goes up, then the measured abundances of all other ASVs will go down. Metagenome contigs have the same problem. This well-known compositionality problem creates spurious negative correlations between features, which in turn could create the illusion of a shared genetic basis where there is none. If the counts were to be corrected for their compositional nature (see e.g., DOI: 10.1093/gigascience/giz107) would these hotspots still be detected?

Thank you for this valuable comment and the appreciation for us having integrated three sets of microbial/plant features. We agree that the non-independent, compositional nature of ASV abundances is a major challenge in microbiome research. In this study, we took this into account and chose to use a computational approach to minimize the bias introduced with Cumulative Sum Scaling (CSS) normalization from the metagenomeSeq package. This computational approach has been shown to correct for several of the biases introduced by total sum normalization. It should be noted that CSS and CLR (centered log ratio, the alternative method referred to by the reviewer) are largely performing similarly, as can be seen in Figure 4b in the recent benchmarking study by the Raes lab (<https://www.nature.com/articles/s41467-021-23821-6>). Nevertheless, experimental correction for total abundance or, preferably, monitoring

ASV abundance quantitatively would be best in future experiments to further minimize the compositionality problem. We have included these considerations in the revised manuscript.

Line 183: “Subsequently, bin and contig abundances were determined by read depth using CSS normalization, a computational method to adjust for compositional bias²⁷”

Line 452: “Here we use CSS normalization, one of the top performing computational approaches to address compositional bias⁶⁹. Nevertheless, future approaches that provide community level absolute ASV abundances will further minimize compositionality of the microbiome data and likely perform better when mapping microbiome features as QTLs.”

Line 572: “Next, ASV counts were normalized using the cumulative sum scaling (CSS), which has been shown to be one of the most effective computational transformation techniques⁶⁹”

I also have a somewhat major critique of the terminology used to describe the significant QTLs and especially their effect sizes. Throughout this paper, each QTL is described as either “wild” or “modern” - in reference to the two parent genotypes of the RIL population. However, in a classic quantitative genetics framework, a QTL is simply a location in a genome where there are 2 segregating alleles with differing effects on the trait of interest; therefore it cannot have the property of “belonging” to one parent or the other. In contrast, each allele at each QTL does come from one of the two parents. Nevertheless, it is not coherent to describe an ASV or other metagenome feature as being linked to either a “wild” allele or a “modern” allele (two examples of many: lines 67-68 and 119). This is because it is the difference between both alleles that defines a QTL. The overall effect of this incorrect terminology is to create confusion about the actual size and direction of the alleles’ effects. In my opinion the simplest and clearest way to fix this problem would be to set one of the two parent alleles as the “reference” and then describe the QTLs not as “wild or modern” but rather “positive or negative” relative to the reference allele. For example, the effect size & direction of the modern allele could be reported for all QTLs. This would be much easier to understand biologically, e.g. it would be easy to understand that an ASV became more abundant or less abundant due to genetic changes during domestication.

Thank you for this valuable comment regarding the terminology. We would first like to note that the effect size and direction for all QTLs were reported, but in retrospect the terms ‘wild’ and ‘modern’ can indeed create confusion. Therefore, we have addressed this concern throughout the revised manuscript by using the modern allele as a reference and describing QTLs for taxonomic and metagenomic features of the microbiome with positive or negative effects relative to the reference allele.

Line 74: Using the modern allele as a reference, we find QTLs for numerous taxonomic and metagenomic features of the microbiome with both positive and negative effects. Interestingly, more positive effects related to increases in microbiome feature abundance were observed for

the modern reference allele compared to the wild reference allele, suggesting that domestication has had a significant impact on rhizosphere microbiome assembly.

Line 116 We identified 48 QTL peaks, across 45 distinct loci, significantly associated with 33 ASVs (Supplemental Table 6). Our logarithm of the odds (LOD) thresholds for significance had been determined by pooled permutations from all ASVs to attain a genome-wide threshold of P 0.05 (LOD 3.35) and P 0.2 (LOD 2.64). The modern allele was set at reference, such that negative effects were relatively more associated with the wild allele and positive effects with the modern allele. Of the significant QTLs, 16 were microbiome features less abundant compared to the reference allele, whereas 32 were microbiome features more abundant in presence of the modern reference allele. The QTLs on chromosomes 11, 10, 8 and 2 were associated with increases in abundance in presence of the modern reference allele. In contrast, the sole QTL on chromosome 7 was negative relative to the reference. All other chromosomes contained a mix of QTLs with positive and negative effects on ASV abundance relative to the reference allele (Figure 3a). While many rhizobacterial lineages were linked to a single QTL (14 out of 25 unique taxonomies), others were linked to two or more QTLs (7 and 4 taxa, respectively) (Figure 3b). Of the lineages with multiple QTLs, most were positive relative to the reference allele. One salient exception was *Methylophilaceae*, with a total of 9 QTLs that were both positive and negative relative to the reference and distributed across chromosomes 3 (positive, x2), 4 (positive), 7 (negative), 11 (positive x2) and 12 (negative x3) (Figure 3c). Another salient feature of the QTL analysis was the hotspot for microbiome assembly identified on chromosome 11, including a significant linkage with ASVs from *Adhaeribacter*, *Caulobacter*, *Devosia*, Rhizobiaceae, *Massilia* and *Methylophilaceae* (Figure 3c).

Line 150: Of further interest is that all diversity metric QTLs were negative relative to the reference

Line 174: While QTLs were identified with both positive and negative effects relative to the reference modern allele, the large number of positive effects suggests domestication impacted rhizosphere microbiome assembly.

Line 374: Numerous *Streptomyces* SNVs were associated positively with the reference tomato alleles on chromosome 6 and 11.

Line 497: In particular, the SNV QTL analysis demonstrated that genes related to the degradation of various plant-associated polysaccharides in *Streptomyces* were associated positively with the modern reference allele

Some more minor comments:

Line 52 “This approach has been adopted” implies that microbiomes are already being used as breeding targets, which as far as I know is not true. In general this paper creates a false

impression that “breeding for the microbiome” is feasible at the scale needed for crop improvement - I suggest adding some caveats about the challenges of knowing which microbiome features to target for crop improvement.

We have altered the sentence as follows:

Line 58: “However, actualizing microbiome features into breeding programs at a scale for crop improvement has not yet been realized. In fact, for most plant species, investigations leveraging diverse plant populations to map microbiome-associated Quantitative Trait Loci (QTL) are still in their infancy^{20,19,18} .

Line 113: What does it mean for a QTL to be “more abundant in” an allele?

Good point; please see above response where this language was revised.

Lines 130-132 and 134: Unclear what effect sizes are being compared

For taxa with multiple QTLs, we were able to statistically compare the effect sizes, showing that the impact of a QTL on the relative abundance of the genus *Massilia* was larger than for other genera. We have adjusted the text as follows:

Line 170: “Collectively, our amplicon analysis provided a broad picture, suggesting that assembly of bacteria in the tomato rhizosphere is a complex trait governed by a combination of multiple loci, some being ASV specific, some being pleiotropic for different ASVs and with heterogeneous effect sizes on ASV abundance (Figure 3d). While QTLs were identified with both positive and negative effects relative to the reference modern allele, the large number of positive effects suggests domestication impacted rhizosphere microbiome assembly.”

”

Lines 153-155: Clarify that the “rhizosphere enrichment” is relative to bulk soil data

We clarified as follows.

Line 189: “With nearly 40 million contigs being assembled, the effects of multiple testing were reduced by prioritizing rhizosphere-enriched contigs (relative to the bulk soil) which were larger than 10kb and with an enrichment greater than 4-fold.”

Lines 151-155: Is there a way to estimate what proportion of the functional diversity was excluded from the metagenome dataset due to these steps? (I am not suggesting that all 40 million contigs should have been tested!)

Thank you for this nice yet challenging suggestion. It is clear that much of the metagenomic data was not included due to our prioritization steps. Statistically, however, it was important to limit the number of tests. Ecologically, we decided it was important to focus on ‘rhizosphere-enriched’ traits only, despite the loss of functional (and taxonomic) diversity that may (or not) harbor other interesting traits. To address your question, we provided an additional analysis to assess the amount of functional diversity, as represented by protein clusters grouped by mutual sequence similarity, which are contained in the rhizosphere enriched contigs. Despite the small

number of contigs, the proteins encoded on these contigs were identified in a rather large number of protein clusters (approximately 8.3% of all protein clusters). Furthermore, the percent of all proteins contained in these clusters was 25%. Thus, while we had strict cut-offs of what was considered rhizosphere enriched (and the number of statistical tests performed was thus effectively reduced to the most relevant ones), there was still a considerable amount of functional diversity encoded by this subset.

Line 192: “The functional potential of these rhizosphere-enriched contigs represented 8.3% of protein clusters identified in all contigs greater than 10kb by MMseqs2 using a 50% protein identity threshold³². Interestingly, approximately 25% of all proteins were contained within these clusters, suggesting that a considerable fraction of functional diversity was maintained during the prioritization. “

Lines 687: “To assess the impact of the prioritization on the functional representation of the metagenome, we identified the fraction of protein clusters represented in the rhizosphere-enriched contigs compared to the rest of the contigs greater than 10kb. First, Prodigal was used in metagenomics mode to predict genes in the metagenomic assembly with contigs longer than 10kbp. Next, MMSeqs2 was used to cluster the protein sequences based on 70% similarity and based on 50% similarity, and with or without partial predicted genes³². To calculate the number of clusters that contained proteins encoded in rhizosphere-enriched contigs, the clusters were searched for the presence of protein IDs of the 1249 rhizosphere-enriched contigs. In total approximately 8.3% of protein clusters contained genes from the rhizosphere-enriched contigs. In addition to proteins contained on rhizosphere-enriched contigs, these clusters contained approximately 25% of all proteins encoded in contigs larger than 10kb (Supplemental Table 20).”

Lines 162-163: QTLs “belonging to” bacterial taxa is confusing - the QTLs are in the plant genome and the contigs are from bacterial genomes. In general, these complex issues require very careful writing, I recommend being very explicit with the language: “QTLs underlying *Streptomyces* contigs”, for example

We have adjusted the language as you suggested.

Lines 208: Similar to above comment: clarify that “root specific genes” are plant genes, not *Streptomyces* genes that are expressed when colonizing the root. Lots of other examples of similar problems

Good point. We have adjusted the text throughout the revised manuscript to avoid confusion.

Line 216: “Of interest” - explain what makes a candidate gene “of interest” for this analysis.

We have removed the term “of interest”.

Lines 237-238: Clarify that if this association is real, the direction of causality is still unknown, the decline in *Cellvibrio* abundance could be a “side effect” of domestication rather than a mechanism of it. The same is true for the interpretation of all other QTL effects

We have adjusted the text as follows

Lines 283: “The QTL on chromosome 1 contains genome-wide sweeps associated with the initial tomato domestication and subsequent improvements of fruit quality traits, suggesting that one or both of these events were connected to or act as a ‘side effect’ on the decreased abundance of *Cellvibrio* in the tomato rhizosphere.”

Line 325: “Streptomyces contig QTLs” → “QTLs for Streptomyces contigs”

Changed in the revised manuscript.

Lines 330-332: An example of where the writing gets confusing because it simultaneously discusses genes in bacteria and plant. For example, clarify “many SNPs” → “many bacterial SNPs” and “chromosomes 6 and 11” → “tomato chromosomes 6 and 11” as often as possible.

Line 376: Numerous *Streptomyces* SNVs were associated positively with the reference tomato alleles on chromosome 6 and 11.

Lines 338-340: This is an interesting observation. In general I really like the attention paid to genetic variation within microbial lineages

Thank you.

Line 349 and several other places: SNV is used instead of SNP to mean the same thing, recommend consistent terminology

The terminology is now consistent throughout the revised manuscript.

Lines 362-365: Reiterate here that functional validation is still required for these candidate genes.

We have explicitly addressed them as ‘putative’

Line 367: “Daunting task” is a better descriptor than “complex phenomenon” in my opinion, as no one is really doing this yet as far as I know

We have changed the languages as suggested. Thank you.

Lines 395-397: Could this discrepancy between amplicon and shotgun data be partially explained by the way most of the contigs had to be discarded, and/or the differences in the completeness of the reference databases used?

Indeed, a good discussion point that is now addressed in the revised manuscript

Line 453: “It is possible that less stringent prioritization steps could be used to increase the number of metagenomic features identified, but this may also increase the false discovery rate.”

Line 409: Fold change compared to what?

We have clarified this point

Lines 451: “The aquaporin (SITIP2.3) has the highest fold change of all tonoplast intrinsic proteins in tomato roots as compared to all other organs^{32,33}, while the FIT gene is a bHLH transcriptional regulator controlling iron homeostasis in tomato^{34,35}.”

Line 449: Were the tomatoes being grown in natural soil, or in commercially-provided potting mix?

We have clarified that this was a natural soil. (**Line 520.**)

Line 465: Covered with what? I assume a transparent lid to allow light?

We have clarified that it was covered with a transparent lid (**Lines 537**).

Line 499: What reference database was used to assign taxonomy?

We have clarified that the database Silva v138 was used to assign taxonomy; for the bulk segregation analysis Silva v132 was used. (**Lines 571 and 765**)

Lines 506+: I appreciate the high level of detail for the metagenomics analyses, but they do seem unbalanced relative to the amount of detail provided for the amplicon data.,

The amount of details provided is necessary for reproducibility. On the one hand, the amplicon data was processed using highly standardized pipelines. In contrast, the computational requirements and complexity of processing the metagenomes required a tailored approach. For example, the assembly of the metagenomics data included multiple assembly strategies that were ultimately merged. Considering the increasing interest in metagenomics, we feel that detailed descriptions of the tailored metagenomics analysis will be instrumental for other future QTL-microbiome analyses involving metagenomics.

Line 631: What exactly was the linear mixed model - for example, which of the predictors were included as random vs. fixed effects?

We have now removed this line from the text as it was confusing. Only the covariates (the number of leaves, harvest day, rhizosphere soil weight (g), soil starting weight (g) and plant dry weight (g)), were added as fixed effects. Please refer to the rQTL manual for additional details, we provide the code used below:

```
out_r <- scan1(genome_probability, phenotypes, addcovar = covariants)
```

Line 636: Permutations of what?

A permutation test was done to determine a significance threshold for QTL.

Line 652: It's unclear what the extracted RNA was used for (as opposed to the DNA used for 16S sequencing)

RNA was not used. Thank you for pointing out this typo.

Reviewer #2 (Remarks to the Author):

the authors used microbiomes of an offspring population of a wild and modern tomato as an external phenotype for QTL analysis. They identified genetic regions that correlated with the association of specific microbes. With metagenome analyses, they additionally identified microbial traits associated with wild or modern tomato QTLs.

the results and figures are presented in a clear, concise way, the manuscript is well written, and the conclusions are supported by evidence. The authors utilized a broad range of modern techniques. Their methodological approach as well as their biological findings will be of interest for a broad readership.

Detailed comments:

L 96/ Fig 1: It is laudable that the authors characterize the RIL population general traits. As these results have been published before and as they do not directly impact the work presented here, I would move this figure to supplemental data.

Thank you for your appreciation of our work. The results of our QTL analyses of the 'classic' plant phenotypic traits are new and they corroborate but also extend previous findings. To provide a solid and reproducible baseline to anchor our microbiome analyses, we think it is crucial to include these data as figure 1 in the core manuscript instead of tucking them away in the supplemental data.

L108/Fig2A: most of the variation is found between bulk soil and rhizosphere samples, which is consistent with other studies. Please also include a PCA plot of rhizosphere samples only to better illustrate differences between the tomato lines.

We have added a PCA plot including only the RIL rhizosphere samples to the supplemental information (Supplemental Figure 3).

L210: please introduce the concept of selective sweeps.

Good point. In the revised manuscript, we have included an additional line introducing the concept of selective sweeps:

Line 251: "61 genes were found in regions previously identified to have selective sweeps

L238: the authors identified quite a number of *Cellvibrio* and *Streptomyces* genes associated with the different QTLs. Are these genes specific to the bacterial strains identified here/ associated with plants or are these common features found in many related strains as well that do not interact with plants?

Great idea to look into this. However, to address this idea properly for all genes involved, one needs to conduct comprehensive database analyses involving comparative genomics on numerous strains, many of which have a poorly documented lifestyle (plant-associated,

rhizosphere, endosphere, ...). In general, though, it is clear that most of these genes do not encode 'housekeeping functions', but functions likely linked to specific niches, such as saprophytism. Some of these niches of course also exist outside the direct association with plants.

L341: please explain the synonymous/ nonsynonymous terminology.

Thank you for this comment, we have included an additional line introducing the terminology of synonymous and non synonymous.

Line 390: "A majority of these SNVs were synonymous having no effect on the produced amino acid sequence. However, some were non-synonymous, resulting in an altered amino acid sequence, including the histidine decarboxylase SNV (B2R_16511) mapping to both tomato chromosomes 6 and 11 (Figure 7)"

best regards, Joelle Schlaepfer

Dear Joelle, thank you for your constructive review. We hope you find our revisions satisfactory.

Reviewer #3 (Remarks to the Author):

In this manuscript, the authors perform QTL mapping in tomato inbred lines, using the root microbiota as the quantitative trait. This innovative method enabled them to reveal new genes in both the plant and the microbiota, involved in microbiome assembly. The combination of both 16S rRNA gene amplicon and metagenomics analysis allowed the authors to map QTLs involved in microbiome assembly both broadly (16S) and relatively precisely (shotgun). This method, facilitated by a wide comparison of inbred lines of a cross between wild and commercial tomato varieties, supplies a thorough insight into tomato genetic basis for microbiome assembly.

This manuscript will be of very high interest to the scientific community and makes an interesting read. I have only minor comments, mostly regarding the presentation of the data, but I also raise some methodological issues. In some parts of the manuscript, a greater attention to detail is needed.

A validation of some of the loci identified using genomic editing in both the plant and the microbiota would have been very welcome, and I hope the authors will perform them down the road, but I think that the amount of work presented is certainly enough for an impactful paper.

Thank you for classifying our manuscript as 'impactful'. Indeed, validation of several loci identified in the plant and the microbiota is on our wish list for the near future.

One important comment that I have regards the paper's take-home message. Many functions are listed, but it is not made clear if how they are all connected and if there is an emerging pattern. Perhaps a summary figure would help illustrate this.

Thank you for this comment regarding a summary figure. We have added figure 8.

The approach the authors take is to consider the relative abundance of each microbial taxon within the microbiota as a quantitative trait. However, this skips over the use of quantitative ecological measures of the community as a whole. Using the methods applied here it should be straightforward to identify QTLs for microbiome alpha and beta diversity (for the latter, a PCoA without bulk soil could be calculated, and then the values along the 1st and 2nd axis could be used as quantitative traits).

Thank you for this comment regarding treating diversity as a microbiome feature to be mapped as a QTL. We have addressed this by analyzing Shannon diversity and PCoA axis 1 and 2.

Line 137: In addition to individual ASVs, we investigated diversity metrics as quantitative traits using Shannon index and Principal Coordinate Analysis (PCoA) with Bray-Curtis dissimilarity. For each approach, we calculated diversity statistics first using all ASVs with a relative abundance greater than the effective samples size²⁷, and second using the rhizosphere-enriched ASVs present in 50% or more of the RIL accessions. For the Shannon index, LOD thresholds for significance were determined by permutations to attain a genome-wide threshold

of P 0.05 (LOD 3.27) and P 0.2 (LOD 2.63). Two QTLs were identified on chromosomes 1 and 3 (Supplemental Figure 1 and 2) using all, and prioritized, ASVs to calculate Shannon Diversity respectively. Of note, the QTL on chromosome 1 overlaps with the confidence interval of the *Cellvibrio* QTL highlighted later in the results section. For the PCoA, the first two components were mapped as quantitative traits. A LOD threshold for significance was determined by permutations to attain a genome-wide threshold of P 0.05 (LOD 3.41) and P 0.2 (LOD 2.71). A single QTL was identified on chromosome 6, interestingly, in the same position as the QTL identified previously for *Streptomyces* ASV 5 (Supplemental Figure 3). Of further interest is that all diversity metric QTLs were negative relative to the reference. Thus, while genetic changes during domestication may have made some ASVs more or less abundant, these genetic changes also impacted overall diversity. Given the non-independence of sequencing-based microbiome features, we suggest caution in interpreting the results of using diversity metrics as microbiome features.

Line 581:To investigate diversity metrics as quantitative traits, the Shannon diversity of each sample was calculated using all ASV after filtering based on the effective sample size using the metagenomeSeq package (v1.28.2)²⁷, and using all ASV in greater than 50% of samples (Supplemental Table 21). Similarly, a PCoA analysis using Bray Curtis distances was conducted, and the values for principle components axis 1 and 2 were extracted (Supplemental Table 22). Both calculations were done in phyloseq version 1.34.0⁸³. These diversity-based microbiome features were then mapped as QTLs as described subsequently.

Another measure that would potentially be very interesting to look at is absolute bacterial abundance. It is likely that there would be QTLs that correlate with the general ability of microbes to colonize the roots. Unfortunately, this manuscript, as many others, does not consider absolute abundance. Perhaps by using the metagenomic data the authors could devise some kind of proxy (e.g the ratio between bacterial and host reads in the data).

Thank you for this comment regarding the absolute abundance, but our data are unfortunately not suitable for this. Nevertheless, we have used the CSS normalization which is the most effective computational method to date to address compositional bias of sequencing data. See also our reply to the comments by reviewer 1 above.

Comments in order of appearance in the text:

Line 18: 16S amplicon is too much of a shorthand in my opinion. I suggest 16S rRNA gene amplicon

We have adjusted the text to read “16S rRNA gene amplicon”.

Line 39: delete “a”

We have made the adjustment.

Line 40: when you say “microbiome breeding programs”, if I understand correctly, you mean plant breeding for specific microbiome selection, rather than breeding of the microbiota themselves. I think that the current wording could be somewhat ambiguous in this respect.

We have made the adjustment to reduce this ambiguity.

Line 55: should be “in their infancy” (I think)

Adjusted.

Figure 1

Here you use either scatter plots or box plots. Why the inconsistency? Elsewhere you use combined box and scatter plots, which are more informative and transparent. Why not use that format for this figure as well? The y-axis height for panels c and d and e and f are not exactly the same.

We have adjusted the boxplots in Figure 1 to include a scatter plot.

“containing neither allele (labeled zero)” this wording is confusing. It is shorthand for “neither BB on 2 and AA on 9”, but they are not referred to yet at this point in the sentence.

We have put this directly into the text to alleviate the confusion.

How many traits were tested in total?

The five covariates were mapped as QTLs.

How does this QTL mapping compare in general to the gnotobiotic QTL mapping? How do the authors interpret contrasting results

In general, multiple corresponding QTLs were identified between our study and previous studies. However, not all QTLs identified in the gnotobiotic experiment were replicated, which is likely due to the differences in experimental conditions.

Line 110: It would be nice to see how abundant and ubiquitous are these ASVs in the data? Perhaps a version of Fig. 2b could be made with these 33 ASVs marked by a different color, and added as a supplementary figure.

Very good point indeed. We have updated figure 2b highlighting the 33 ASV with QTLs.

Line 117: Here it says 14 taxa out of 25 but above in line 110 you say you found 33 ASVs. What do you mean here by “taxa” and how does this square with the number of ASVs? From the taxonomic names in Figure 3 I understand that abundance at all taxonomic ranks was considered, but I could not find this explained anywhere. Perhaps I missed it, but this could be clarified better here.

Many of the ASV belong to the same taxonomic classification. Hence, while there are 33 unique ASV, there are only 25 unique taxonomies. We have adjusted the text to clarify this.

Line 126: While many rhizobacterial lineages were linked to a single QTL (14 out of 25 unique taxonomies), others were linked to two or more QTLs (7 and 4 taxa, respectively) (Figure 3b).

Figure 3:

Panels are labeled with uppercase letters in the text and lowercase letters in the figure (here and elsewhere).

We have made this more consistent.

The effect size units in panel C could be more explicit. Perhaps explain this more in the legend. It is not trivial enough to just be denoted by a % sign

Figure 3d* panel text: “Effect size was calculated as the percent change relative to the mean CSS abundance for each ASV.” (*note the panels were re-arranged as suggested subsequently.)

(on a related note - why is it important to statistically compare effect sizes among taxa? The rationale for this analysis is not made clear in lines 125-140).

We have included a more explicit definition of the effect size in the legend as suggested. We have also included additional background as to why we were interested in comparing the effect size between taxa (it is not trivial that the effect size should be different between taxa, this is the first time these types of statistics are reported. Here we show that the effect sizes differ between QTL and between taxa).

Line 158: “Effect size is an important factor when mapping the genetic architecture of quantitative traits. While some QTLs have large effect sizes, many small effect QTLs may explain a large proportion of trait variation²⁸. To date, there is little understanding of the distribution of the effect sizes of QTLs for microbiome features. Here we show that the absolute values of the effect sizes of the 48 QTLs on ASV relative abundance ranged from 1.3 to 17%, with an average effect size of approximately 5%, comparable to the effects seen for SDW and RM (Figures 1c and 1e).”

Also, the text refers first to panel D and then to panel C. That’s confusing.

Panels c and d were switched.

The color shades of panels B and C don’t match. What are the colors there for anyway?

The colors were adjusted and are now consistent. When color is not necessary, it was removed.

Panel D:

You do not explain what the edge colors denote. I assume they correspond to the wild/modern colors in panel A but the reader does not have to figure that out on their own. Also the color shades don’t match. In addition, some of the nodes seem to change color when passing through the edges. The figure caption ends with the statement that “A complex network emerges”. This is a rather diffuse statement. Complex in relation to what? Much of the complexity of this network results from the choice to include multiple taxonomic ranks in preparing it. Moreover, many data structures can be presented in a network form, but that does not suffice to conclude

that this is a truly interacting network as implied here. All in all I get the impression that this figure panel was included because it looks cool (it really does) but that the authors struggle to draw a meaningful conclusion from it.

The color was changed to match. The edges no longer change color when passing through nodes. The phrase “a complex network emerges was removed”.

Line 145: looks like one sample was switched in error (no need to make any changes here, these things happen...).

Correct.

Line 148: At least some readers will find this confusing. I think a sentence would be in place here to say that bin and contig abundances were determined by read depth? I know it appears in the Methods, but please spell out CSS here. It is a bit unbalanced to rush through the normalization and abundance calculations here, but on the other hand devote a sentence to list which software was used for binning. In general, I think some of the details in the Methods need to be included here, like the fact that you devised a strategy to co-assemble ALL metagenomes, allowing you to map reads from all samples to the contigs and perform a differential abundance analysis. I think that a reader that skips looking at the methods should still be made aware if this A1-A2-A3 strategy.

Thank you, we have made the recommended adjustments.

Line 181: “After pre-processing, a co-assembly strategy using all metagenomes was implemented (see Supplemental Methods section 4.2.2 for more detail). Subsequently, bin and contig abundances were determined by read depth using CSS normalization, a computational method to adjust for compositional bias.”

Line 151: Many open questions here: how do the taxonomic profiles from metagenomics and 16S compare? Do the soil, MoneyMaker and wild samples separate on a PCoA space similarly as the 16S data?

We appreciate the curiosity of the reviewer. Although we cannot address all these questions, we did include a PCoA of the metagenomics data in the revised manuscript. (Supplemental Figure 9)

How many of the contigs were not bacterial? How were these used downstream?

The taxonomy of the rhizosphere enriched contigs was determined using Kraken and there were no non-bacterial QTLs identified. A comprehensive analysis of all the contigs was not done, nor is it necessary for the approach taken.

Line 152: “we took numerous ...” this is an odd choice of words. I can’t imagine you used so many prioritization steps that you couldn’t count them. Can’t you just say how many steps? Good point. The steps are explained in the next sentences. We added the lines “as described subsequently”.

Figure 4: See comments for Figure 3. Also, I understand why you use a log scale here in panel B but I cannot understand why the boxes are all different sizes. Very confusing. Isn't each box one contig QTL? What is count+1,2?

We recognize this may be confusing, but the count data was normalized by $\log_2(x+1)$ transformation. The boxes are different sizes because they represent a different number of contig QTLs. The larger the box, the more contig QTLs were identified.

We made the adjustments suggested.

Figure 5

What are the units of the y-axis? The legend says relative abundance? So why are the values so high?

The legend has been changed in the revised manuscript to say normalized CSS abundance.

Why do the colors signify? Seems like they have no meaning, in which case I suggest removing them. This is doubly confusing considering a similar palette was used to signify something else in previous figures.

The color was removed.

It does not say anywhere what M and P stand for. I assume these are the parental lines? M for Moneymaker/Modern? P for pimpinellifolium? I don't get it.

The legend has been clarified in the revised manuscript.

Figure 6

This figure sends the reader to do too much homework. y-axis should say $\log_2(\text{root/leaf expression ratio})$. The legend title for panel b should not be "Legend (FPKM)". You could change this to "Transcript abundance ($\log_2(\text{FPKM})$).". You do not spell-out FPKM anywhere.

The legend has been clarified in the revised manuscript.

Line 206: RNA-seq

Thank you.

Line 236: it is not entirely clear what the authors think is the relationship between these genome-wide sweeps and the microbiota are. Is it causal, or more coincidental? Could the authors try and explain this and the evidence supporting one or the other more precisely?

Good point indeed. We do not know yet whether the relationship is causal or coincidental; we demonstrate that the selective sweeps are in regions related to microbiome assembly showing a clear link (for the first time) that the domestication process impacted alleles involved in microbiome assembly. We have clarified this in the text.

Lines 254: "While it remains unclear whether the relationship between selective sweeps and changes in microbial feature abundance is causal or coincidental; here we demonstrate for the

first time a clear link and genomic signature that the domestication process impacted alleles involved in microbiome assembly.”

Lines 304-309. I completely agree. In a previous paper (Carrion et al, 2019) the same authors actually took this step and validated metagenomic hypotheses by site-directed mutagenesis. Here such validations can be done with genomic editing in both bacteria and plants, which would substantially elevate the impact of this manuscript. I am not suggesting however that such experiments should be a prerequisite for publication in my opinion.

We indeed hope we can successfully validate these hypotheses using site-directed mutagenesis in the identified microbial taxa as well as in the host plant.

Line 365: This manuscript unloads a huge amount of information on the reader. The authors did a good job at highlighting in the text numerous functions in both the plant and the microbiome that the analysis identified. Some kind of synthesis is needed here, to highlight the patterns that emerge. The take-home sentence at the end of the Results section focuses on the method itself rather than the findings. I think that this is a missed opportunity. A graphical summary of the findings would go a very long way here to highlight the patterns of plant-microbe interactions that emerge from this study.

See figure 8.

Lines 396-397: Can you provide an explanation for this tradeoff? Is it because metagenomics goes into a deeper taxonomic resolution while amplicon allows us to see less abundant taxa? That seems like the obvious answer but I think you should spell it out.

Very good point indeed. We added this to the revised manuscript.

Lines 438: “Amplicon-based sequencing, which captures more rare taxa per unit sequencing, provided a broader taxonomic picture and was able to capture QTLs of both abundant and relatively rare rhizobacterial lineages.”

Line 405: “driving forces driving” change to “forces driving”.
corrected

Methods:

This paper employs many state-of-the-art methods. I suggest that to make things easier on the reader, the authors add to each sub header which section in the Results and which Figure this section refers to.

The current manuscript structure is well-sectioned and already contains headers and two levels of sub headers (e.g. 4.1.1). We experimented with additional sub headers cross referencing the headers as suggested, however we felt these detracted from readability. We have therefore we have maintained the current structure.

Line 499: Supplementary table 2 still contains the mitochondrial and chloroplast reads. On that matter, the supplementary tables are provided with no caption or explanation and with meaningless file names. Please amend that. In addition, in the text, (Line 100) you refer to sup. Tables 2-5 in bulk, without explaining what each one is.

We have removed this line and now reference each supplemental table 2-5 separately in section 4.2.1 (rRNA amplicon sequence processing)

Line 516: isn't there a sentence missing here saying something along the lines of "then we mapped all reads from the 96 RIL samples to this assembly"?

corrected

Line 564: "genomic genomics"

corrected

Line 606: here and in the corresponding Results section, can you explain how SNPs and SNVs relate to one another?

Regarding the use of SNV and SNP, we have decided to solely use SNV. See also reply to reviewer 2 above

There are a few other things here that I find confusing. What are the "1249 contig enriched genomes"? In an earlier paragraph you say that you identified 1249 rhizosphere enriched contigs. So is this SNP analysis done on contigs or on genomes?

Or on MAGs containing these 1249 contigs? I'm confused.

Apologies, this was a very odd typo. The analysis was done on the rhizosphere-enriched contigs.

Furthermore, in either case, how does SNP calling work when you are looking at a diverse collection of genomes /contigs not necessarily belonging to a single lineage? Do you only consider core orthologous genes? Do you map reads to these contigs to identify any polymorphisms among the mapping data? If so, is it the same mapping as before or do you re-map the reads? Please add some more details to explain the analysis pipeline better.

We used the previously published tool inStrain for this analysis. Please refer to the inStrain manuscript (<https://doi.org/10.1038/s41587-020-00797-0>) for more details about the method. They employ a complex algorithm to identify SNVs from metagenomes in a microdiversity-aware manner.

Reviewer #4 (Remarks to the Author):

The manuscript submitted by Oyserman and colleagues presents a beautiful work on investigating the genetic basis of rhizosphere microbiome assembly in tomato. Manuscript is well written and clearly structured. Authors provided solid data in both host plant and the microbiome. Main conclusions can be supported by the provided evidence. I agree that this

study shed light on a new approach for further understanding plant microbiome interactions and will be surely benefit for plant microbiome breeding program.

Thank you for your enthusiasm and constructive comments

Specific comments:

1. Except for QTLs identification, it is worth to check phenotypical variation of the RIL populations compared to modern *Solanum lycopersicum* var. MoneyMaker and wild *Solanum pimpinellifolium*. Linking genetic architecture with phenotypic observations and the microbiome assembly will make the investigations of this study more powerful and meaningful.

Linking genetic architecture, microbiome composition, with detailed above and belowground phenotyping will be the subject of future experiments in which the specific effects of microbiome members on plant phenotypes will be part of future validation experiments.

2. Both wild and modern alleles were identified, large number of modern alleles were identified with highly associations of rhizosphere microbiome assembly. I wonder what if this experiment was conducted under a field condition with natural soil associated with local microbiomes. In this situation, I would expect larger number of wild alleles than modern will be identified with highly association of microbiome assembly. So, did you perform further analysis to prove your conclusions with any soils which differs to the commercial greenhouse soil?

Thank you for this great suggestion for future studies. We expect that working with native soils with other microbial taxa and microbial functions may highlight additional associations in addition the ones identified in this study.

3. Rhizosphere microbiome composition is closely related to root exudates. Compared to bulk soil communities, rhizosphere reflects higher plant hosts selection that influence microbiome assemblies. Considering rhizosphere effects, compared to the rhizosphere, endophytic compartment will have even stronger plant host selection. Plant genetic variation has stronger effects on microbiome assembly in the endophytic compartment. Please discuss why in this study authors considered focusing on rhizosphere microbiome instead of microbiomes in endophytic compartment.

Very good point that the endophytic compartment would also be interesting to include. We did consider this but as we cannot yet properly separate the endophytic microorganisms from the plant cells, in-depth shotgun metagenomics on the endophytic microbiome was not possible. Therefore, we decided to focus on the rhizosphere.

4. Validation of *Cellvibrio* and *Streptomyces* 16S rRNA QTLs with bulk segregant analysis showed in Fig.5. This independent experiment was conducted with modern, wild and 77 RIL accessions. Why didn't use all 96 RIL accessions? Please further clarify it.

Obtaining sequencing data for all 96 RIL accessions was challenging in this experiment due to not sufficient DNA or incomplete set of replicates for a specific RIL accession. Nevertheless, the 77 RIL accessions that did qualify are more than sufficient for the bulk segregant analysis.

Reviewers' Comments:

Reviewer #1:

Remarks to the Author:

I am satisfied with the authors' responses to my comments on the original manuscript, and to the other reviewers' comments. I have no further suggestions.

Reviewer #2:

Remarks to the Author:

Dear authors,

I only have one minor alteration after this first review process: for Supplemental Figure 3, please indicate RIL rep1, rep2, modern and wild cultivars, as shown in figure 2A. Else, the PCA plot is not very informative.

Aside from updating this figure, I am satisfied on how my comments were addressed.

Reviewer #3:

Remarks to the Author:

Thank you for your detailed and thorough response to my comments. With regards to my comment on the diversity metrics, the authors now write that "diversity metric QTLs were negative relative to the reference". Are the authors referring to the PCoA axes as diversity metrics? If so, then I do not understand what "negative relative to the reference" means in this context, since the direction of the PCoA axis is meaningless. If this is just my misunderstanding, feel free to ignore.

Otherwise I have no additional comments and I endorse publication!

Reviewer #4:

Remarks to the Author:

The revised manuscript by Oyserman et al. addressed most of my concerns satisfactorily. Only several minor points need to be further discussed.

New comments

1. According to my understanding, results in 2.1 were used to prove the reproducibility of authors' QTL mapping method. The plant traits dry weight and rhizosphere mass were not used in subsequent analysis. To avoid ambiguity, the relationship between this part and the rest of the study needs to be further clarified. Reviewer 2 also had similar concern.
2. In the manuscript, the authors claimed that 48 QTLs, across 45 loci, significantly associated with 33 ASVs were identified. Only *Cellvibrio* and *Streptomyces* were selected for further investigation. Without experimental validation, please further clarify the process and criteria for finding out these two specific bacteria.
3. Line 108-109: should be "16S rRNA gene amplicon sequencing" and many other places in the main text, the authors use 16S to represent 16S rRNA gene.
4. Line 531: The natural soil was not used in this study, please correct it.
5. Line 676: 18S? the length of V3-V4 region (341-805) is longer than 300bp. How can the authors

use PE150 to get the full-length V3-V4 data? PE250?

6. Many parts of the method are redundant, the authors should re-organize them. For example, the authors probably can combine 4.2.1 and 4.3.

REVIEWERS' COMMENTS

Reviewer #1 (Remarks to the Author):

I am satisfied with the authors' responses to my comments on the original manuscript, and to the other reviewers' comments. I have no further suggestions.

Thank you for providing the valuable feedback

Reviewer #2 (Remarks to the Author):

Dear authors,

I only have one minor alteration after this first review process: for Supplemental Figure 3, please indicate RIL rep1, rep2, modern and wild cultivars, as shown in figure 2A.

Else, the PCA plot is not very informative.

Aside from updating this figure, I am satisfied on how my comments were addressed.

Thank you for the valuable feedback. Color has been added to indicate RIL replicates 1 and 2. Modern and Wild cultivars were not included as they are not part of the QTL analysis.

Reviewer #3 (Remarks to the Author):

Thank you for your detailed and thorough response to my comments. With regards to my comment on the diversity metrics, the authors now write that “diversity metric QTLs were negative relative to the reference”. Are the authors referring to the PCoA axes as diversity metrics? If so, then I do not understand what “negative relative to the reference” means in this context, since the direction of the PCoA axis is meaningless. If this is just my misunderstanding, feel free to ignore.

Otherwise I have no additional comments and I endorse publication!

Here we use Shannon diversity as a diversity metric and are referring to this, not the PCoA. Kind regards and thank you for the valuable feedback.

Reviewer #4 (Remarks to the Author):

The revised manuscript by Oyserman et al. addressed most of my concerns satisfactorily. Only several minor points need to be further discussed.

We highly appreciate the considerable attention you have given this manuscript.

New comments

1. According to my understanding, results in 2.1 were used to prove the reproducibility

of authors' QTL mapping method. The plant traits dry weight and rhizosphere mass were not used in subsequent analysis. To avoid ambiguity, the relationship between this part and the rest of the study needs to be further clarified. Reviewer 2 also had similar concern.

The plant dry weight and rhizosphere mass were used as covariates in the QTL analysis. We indicated this at several points in the revised manuscript (Lines 136, 333 & 1157).

2. In the manuscript, the authors claimed that 48 QTLs, across 45 loci, significantly associated with 33 ASVs were identified. Only *Cellvibrio* and *Streptomyces* were selected for further investigation. Without experimental validation, please further clarify the process and criteria for finding out these two specific bacteria.

Good point. The clarification is explicitly mentioned in the revised manuscript: see for example line 539: "The two most abundant rhizosphere taxa with replicated patterns for amplicon and metagenome-based QTLs were *Streptomyces* and *Cellvibrio*."

3. Line 108-109: should be "16S rRNA gene amplicon sequencing" and many other places in the main text, the authors use 16S to represent 16S rRNA gene.

Thank you for noticing. This change has been made throughout the text.

4. Line 531: The natural soil was not used in this study, please correct it. We indicated the origin of the soil and how we processed it in this study. We used the term 'natural' to differentiate from studies that are increasingly using synthetic soils. Moreover, the word 'natural' was added on request of another reviewer to make this distinction.

5. Line 676: 18S? the length of V3-V4 region (341-805) is longer than 300bp. How can the authors use PE150 to get the full-length V3-V4 data? PE250?

The sequencing strategy used 300 bp paired end sequencing (see line 982).

6. Many parts of the method are redundant, the authors should re-organize them. For example, the authors probably can combine 4.2.1 and 4.3.

We appreciate your suggestion, however, we prefer to give considerable attention to describe the methods in a concise, precise and complete manner. Re-organizing or combining sections will affect the level of precision that is needed.